# Spectra: Rethinking Optimizers for LLMs Under Spectral Anisotropy

Zhendong Huang [* 1]  Hengjie Cao [* 1 2]  Fang Dong [1]  Ruijun Huang [1 3]  Mengyi Chen [1]  Yifeng Yang [1]
Xin Zhang [1]  Anrui Chen [1]  Mingzhi Dong [4]  Yujiang Wang [5 6]  Jinlong Hou [2]  Qin Lv [7]  Robert P. Dick [8]
Yuan Cheng [2]  Tun Lu [1]  Fan Yang [1]  Li Shang [1 9]

## Abstract

Gradient signals in LLM training are highly anisotropic: recurrent linguistic structure concentrates energy into a small set of dominant spectral directions, while context-specific information resides in a long tail. We show that this spike–tail separation persists throughout training, with the spike occupying only about $1.5\%$ of directions yet dominating optimizer statistics. This dominance suppresses tail learning by contracting tail updates through second-moment normalization and tightening the globally stable learning-rate bound. Motivated by this analysis, we propose *Spectra*, a spike-aware optimizer that suppresses the dominant low-rank spike subspace without amplifying the noise-sensitive spectral tail. Spectra tracks the spike subspace via cached, warm-started power iteration and applies low-rank spectral shaping with negligible overhead and substantially reduced optimizer-state memory. Across Qwen3-0.6B trained on 100B tokens and LLaMA3-8B trained on 50B tokens, Spectra achieves the lowest final validation loss, improving average downstream accuracy by 1.41/0.89 and 1.62/0.66 points over AdamW/Muon, respectively. For wall-clock convergence, Spectra reaches matched loss targets up to $1.31\times$, $1.34\times$, and $1.24\times$ faster than AdamW on Qwen3-0.6B, Qwen3-2B-A0.8B, and Qwen3-8B; its speedup over Muon grows as model scale increases from 0.6B to 8B. For computational efficiency, Spectra is $5.1\times$ faster than Muon

in optimizer processing time, cutting optimizer-state memory by $49.25\%$ and achieves the lowest measured end-to-end per-step runtime . Spectra's Megatron integration is available at https://github.com/kimmichtank/spectra.

## 1. Introduction

Training on natural language corpora produces a highly imbalanced learning signal: grammatical and functional patterns recur ubiquitously across data, whereas semantically rich world knowledge is distributed over a vast long tail of rare and sparsely sampled events (Piantadosi, 2014; Linders & Louwerse, 2023; Mikhaylovskiy, 2025). This imbalance induces strong directional correlations in representation space, resulting in highly anisotropic contextual embeddings (Arora et al., 2017; Mu et al., 2017; Ethayarajh, 2019; Li et al., 2020; Timkey & Van Schijndel, 2021). Consequently, gradients associated with common linguistic structure are repeatedly reinforced during training, while gradients corresponding to long-tail semantic content remain weaker, and more intermittent (Kandpal et al., 2023).

This work aims to characterize the anisotropic structure of gradient signals in LLM training and translate it into optimizer design principles. Our analysis is grounded in a spectral perspective because the imbalance is *directional rather than element-wise*: individual coordinates entangle multiple latent factors and obscure independent learning modes, whereas spectral analysis reveals how skewed signals concentrate into a small set of correlated directions (Cao et al., 2025; Ethayarajh, 2019; Timkey & Van Schijndel, 2021). Our analysis yields four key observations:

**Observation 1: A common-structure low-rank spike with a smooth semantic tail.** The gradient spectrum exhibits a pronounced low-rank spike: roughly the top $1.5\%$ directions carry a disproportionate fraction of gradient energy and are separated from the tail spectrum by one to two orders of magnitude. This anisotropy phenomenon is consistent across model scales, modules, and training stages, and admits a two-region semantic correspondence: the spike is primarily driven by common linguistic structure, while the

*Equal contribution  [1]Fudan University, Shanghai, China
[2]Shanghai Innovation Institute, Shanghai, China [3]Research Institute of Tsinghua University in Shenzhen, Shenzhen, China
[4]University of Bath, Bath, United Kingdom [5]Department of Engineering Science, University of Oxford, Oxford, UK [6]Oxford Suzhou Centre for Advanced Research, Suzhou, China [7]University of Colorado Boulder, Colorado, USA [8]University of Michigan, Michigan, USA [9]Shenzhen Loop Area Institute, Shenzhen, China. Correspondence to: Li Shang <lishang@fudan.edu.cn>.

*Proceedings of the $43^{rd}$ International Conference on Machine Learning*, Seoul, South Korea. PMLR 306, 2026. Copyright 2026 by the author(s).

tail encodes finer, context-specific semantic variations.

**Observation 2: Spike updating suppresses long-tail learning.** Spike directions dominate AdamW's (Kingma, 2014) second-moment accumulation, so element-wise normalization is effectively set by the spike subspace and contracts tail update magnitudes. Moreover, spike-dominated gradient variance bounds the optimal learning rate, imposing a conservative global step size that further limits progress in long-tail directions.

**Observation 3: Smaller singular directions carry sparser semantics and higher statistical relative variance.** Along the spectral tail, semantic signals become increasingly sparse and intermittent: only a diminishing fraction of samples yield non-negligible projections onto smaller-singular directions. Accordingly, their relative variance rises sharply, meaning stochastic fluctuations dominate as singular values decrease. As a result, updates in these small-singular directions are increasingly unstable and easily drowned out.

**Observation 4: Numerical variance further destabilizes the spectral tail under iterative updates.** Spectrum-aware optimizers, such as Muon (Jordan et al.), often implement spectral processing via iterative routines, such as Newton–Schulz iteration (Higham, 1997), which introduce numerical perturbations that concentrate in small-singular components, amplifying tail disturbances; aggressive tail equalization further worsens this while adding substantial computation.

**Spectra: design and properties.** Our analysis motivates a clear design principle: *suppress the dominant low-rank spike subspace while avoiding aggressive amplification of the fragile spectral tail*. Guided by this principle, *Spectra* is designed with the following properties:

*Efficiency.* Spectra tracks only the low-dimensional spike subspace using intermittently updated, warm-started power iteration and operates exclusively on this fixed small-rank component. As a result, it incurs low computational overhead and avoids storing per-parameter second-order statistics, substantially reducing optimizer-state memory.

*Optimization.* By selectively attenuating spike-dominated updates, Spectra prevents common features from dominating optimization dynamics and avoids amplifying noise-dominated spectral components. This relaxes the spike-induced gradient-variance constraint on stable learning rates, widening the effective learning-rate range and accelerating convergence.

*Parallelism.* Low-rank spike subspace estimation is naturally distributed-friendly: power iteration can be implemented via local GEMMs with only lightweight collectives on low-rank quantities. This makes Spectra well suited for large-scale parallel and distributed training without requiring full-gradient synchronization.

We evaluate *Spectra* on Qwen3-0.6B trained on 100B tokens and LLaMA3-8B trained on 50B tokens, and further measure wall-clock and per-step efficiency across additional Qwen3 scales.

**Convergence and downstream performance.** Spectra achieves lower final validation loss than both AdamW and Muon on Qwen3-0.6B and LLaMA3-8B. On Qwen3-0.6B, it improves average downstream accuracy by $+1.41$ and $+0.89$ points over AdamW and Muon, respectively. On LLaMA3-8B, it improves average accuracy by $+1.62$ and $+0.66$ points over AdamW and Muon, respectively.

**Wall-clock convergence.** In time-to-target evaluations, Spectra reaches matched loss levels up to $1.31\times$, $1.34\times$, and $1.24\times$ faster than AdamW on Qwen3-0.6B, Qwen3-2B-A0.8B, and Qwen3-8B, respectively. Its advantage over Muon also increases with scale, indicating that the benefit of localized spectral suppression becomes more pronounced at larger model scales.

**Computational efficiency.** Spectra is $5.1\times$ faster than Muon in optimizer processing time, cutting optimizer-state memory by $49.25\%$, and in our Qwen3 runtime benchmarks from 0.6B to 32B, it yields lower measured end-to-end per-step runtime than both AdamW and Muon.

## 2. Analysis

### 2.1. Gradient Anisotropy: A consistent characteristic

For a gradient matrix $\mathbf{G} \in \mathbb{R}^{m \times n}$, Singular Value Decomposition (SVD) is applied to obtain singular values $\{\sigma_i\}_{i=1}^{\min(m,n)}$, left (right) singular vectors $\{\mathbf{u}_i\} \in \mathbb{R}^m$ ($\{\mathbf{v}_i\} \in \mathbb{R}^n$), such that $\mathbf{G} = \sum_{i=1}^{\min(m,n)} \sigma_i \mathbf{u}_i \mathbf{v}_i^\top$. We assume singular values are sorted in descending order, i.e., $\sigma_1 \geq \sigma_2 \geq \cdots \geq \sigma_r \geq 0$ with $r = \min(m, n)$.

Across Qwen3 (Yang et al., 2025) models of different scales, from 0.6B to 32B, and at different training stages, Figure 1 reports the gradient singular spectrum of the deepest MLP layer. Results for attention modules and shallower layers are provided in Appendix A.1. In all cases, *the gradient spectrum exhibits a consistent anisotropic pattern*, following a "low-rank spike + smooth tail" profile: a compact spike block is separated from the tail by roughly one to two orders of magnitude in singular value. In practice, the spectral region preceding the first eigengap occupies a small and stable fraction of directions, approximately $1.5\%$ across model scales, we therefore adopt this value as a stable default.

### 2.2. Linguistic Correspondence of Gradient Anisotropy

This subsection attributes gradient anisotropy to different linguistic signals in the training data. Specifically, we analyze gradient spectra on Qwen3-0.6B (Yang et al., 2025) and

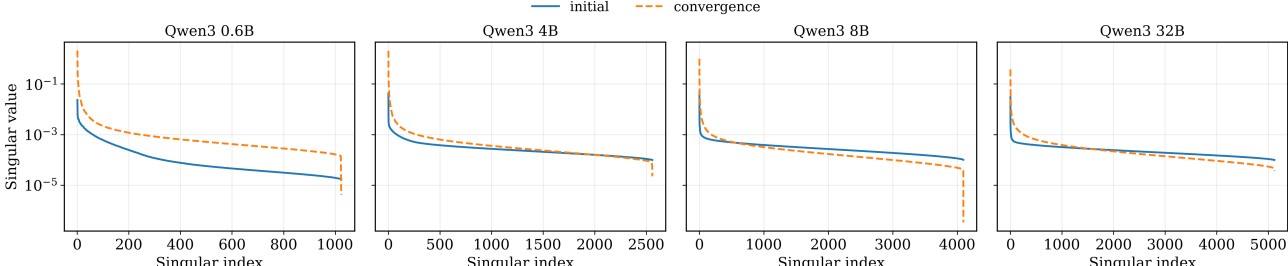

*Figure 1.* Singular-value spectra of the deepest-layer MLP gradient in Qwen3 models (0.6B–32B) at multiple training stages exhibit a consistent "low-rank spike + smooth tail" profile, with spike singular values separated from the tail by ∼1–2 orders of magnitude and occupying a nearly constant ≈ 1.5% of directions.

LLaMA3-8B (Dubey et al., 2024). For each model, we compute the gradient matrix $\mathbf{G}$ under three controlled conditions: (i) *Raw*, serving as the unmodified reference; (ii) *FreqNorm*, reducing the excessive contribution of high-frequency tokens; and (iii) *Shuffle*, removing syntactic dependencies and sequential structure. For *FreqNorm*, with token $t_j$ at position $j$ and corpus frequency $f(t_j)$, we rescale the token-wise loss as $\tilde{\ell}_j = \ell_j / f(t_j)$ and backpropagate from $\sum_j \tilde{\ell}_j$. For *Shuffle*, we randomly permute tokens within each sentence before computing gradients. We then compare the singular spectra of $\mathbf{G}$ across conditions to localize which spectral regions are sensitive to frequency skew and syntactic-order perturbations.

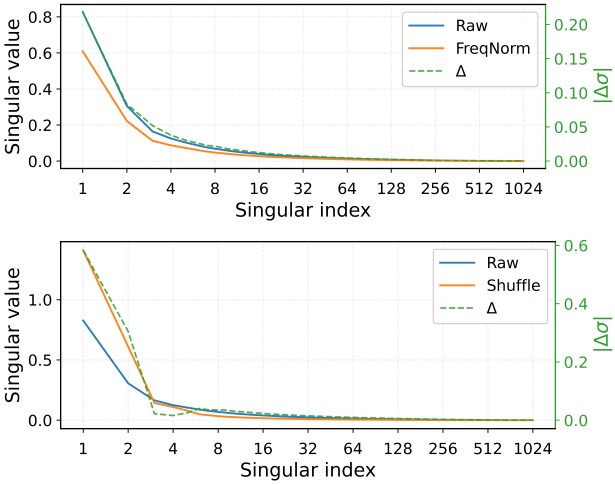

*Figure 2.* Gradient spectrum under two controlled interventions on Qwen3-0.6B: frequency-normalized loss (*FreqNorm*, top) selectively suppresses the leading spike components, while intra-sentence token permutation (*Shuffle*, bottom) selectively amplifies them; in both cases, changes rapidly vanish in the tail.

Figure 2 reports results on Qwen3-0.6B, with LLaMA3-8B deferred to Appendix A.2, and shows that both interventions perturb the spectrum almost exclusively in the spike: the absolute change $|\Delta\sigma_k|$ concentrates on the leading singular components and quickly vanishes in the tail. Under *FreqNorm*, the spike is selectively suppressed and the largest

singular value drops by more than 25%, indicating that a substantial fraction of spike energy is driven by high-frequency-token contributions. Under *Shuffle*, the spike is selectively amplified, leading components increase by up to 75%, because disrupting word order removes syntactic structure that the pretrained model expects, inducing large corrective gradients concentrated in the spike subspace.

Together, these responses indicate that *the spike predominantly reflects common grammatical signals driven by frequency skew and order-sensitive structure, while the smooth tail is comparatively robust and more associated with fine-grained, context-dependent semantics.*

### 2.3. Spike Updating Suppresses Long-Tail Learning

This subsection shows that spike dominance suppresses long-tail learning through two coupled mechanisms. First, under AdamW-style optimization, spike-dominated second-moment accumulation controls element-wise normalization and contracts tail update magnitudes. Second, spike-dominated stochastic gradient variance imposes a conservative learning-rate ceiling.

We measure the spectral structure of AdamW momentums on Qwen3-0.6B and LLaMA3-8B during pretraining. We record the first moment $\mathbf{M}$ and the second moment $\mathbf{V}$, and compute their singular spectra together with the cumulative energy distribution (CDF), $\text{CDF}(j) = \sum_{i=1}^{j} \sigma_i^2 / \sum_{i=1}^{r} \sigma_i^2$. To isolate spike-dominated normalization, we decompose each moment into a spike projection and a residual tail: $\mathbf{M} = \mathbf{M}_s + \mathbf{M}_t$ and $\mathbf{V} = \mathbf{V}_s + \mathbf{V}_t$, where $\mathbf{M}_s \triangleq P_k(\mathbf{M})$ and $\mathbf{V}_s \triangleq P_k(\mathbf{V})$ denote the rank-$k$ truncated SVD reconstructions, and $\mathbf{M}_t \triangleq \mathbf{M} - \mathbf{M}_s$, $\mathbf{V}_t \triangleq \mathbf{V} - \mathbf{V}_s$. Under the AdamW update $\Delta\mathbf{W} = -\eta\,\mathbf{M}/(\sqrt{\mathbf{V}} + \epsilon)$, the tail contribution is $\Delta\mathbf{W}_t = -\eta\,\mathbf{M}_t/(\sqrt{\mathbf{V}_s + \mathbf{V}_t} + \epsilon)$, showing that tail updates are normalized by a denominator dominated by $\mathbf{V}_s$ when $\mathbf{V}$ is highly anisotropic. We visualize the suppression by comparing the element-wise magnitudes of $\mathbf{M}_t/(\sqrt{\mathbf{V}_s + \mathbf{V}_t} + \epsilon)$ against the tail-only baseline $\mathbf{M}_t/(\sqrt{\mathbf{V}_t} + \epsilon)$.

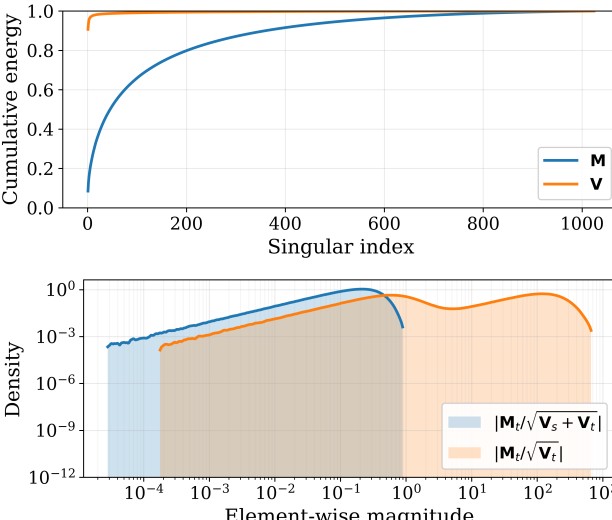

*Figure 3.* Spike-dominated second-moment accumulation suppresses tail updates (Qwen3-0.6B). *Top:* cumulative spectral energy (CDF) of AdamW moments, showing that the second moment $\mathbf{V}$ is far more spike-concentrated than the first moment $\mathbf{M}$. *Bottom:* element-wise magnitudes of tail updates, where full normalization $\mathbf{M}_t/(\sqrt{\mathbf{V}_s + \mathbf{V}_t} + \epsilon)$ is strongly contracted relative to the tail-only baseline $\mathbf{M}_t/(\sqrt{\mathbf{V}_t} + \epsilon)$.

Figure 3 (Qwen3-0.6B) shows that AdamW's second moment $\mathbf{V}$ is much more spike-dominated than the first moment $\mathbf{M}$: the spike subspace already explains about 97% of the spectral energy in $\mathbf{V}$, while accounting for only about 50% in $\mathbf{M}$. Results on LLaMA3-8B are provided in Appendix A.3. This separation implies that element-wise normalization is effectively governed by spike-driven variance accumulation. Accordingly, the tail-update distribution under full normalization, $\mathbf{M}_t/(\sqrt{\mathbf{V}_s + \mathbf{V}_t} + \epsilon)$, is markedly smaller than the tail-only baseline $\mathbf{M}_t/(\sqrt{\mathbf{V}_t} + \epsilon)$, with the latter shifted upward by roughly two orders of magnitude. Overall, spike-dominated second-moment accumulation sharply contracts the effective step size available to long-tail directions.

Beyond element-wise suppression, spike-dominated stochastic variance also bounds the mean-optimal learning rate. Consider the expected loss $L(\mathbf{w})$ for a layer weight matrix $\mathbf{W} \in \mathbb{R}^{m \times n}$ with $\mathbf{w} \triangleq \operatorname{vec}(\mathbf{W})$. Let $\mathbf{G} \in \mathbb{R}^{m \times n}$ be the random mini-batch gradient computed with batch size $B$, with mean $\bar{\mathbf{G}} \triangleq \mathbb{E}[\mathbf{G}]$. Vectorizing gives $\mathbf{g} \triangleq \operatorname{vec}(\mathbf{G})$ and $\bar{\mathbf{g}} \triangleq \mathbb{E}[\mathbf{g}] = \operatorname{vec}(\bar{\mathbf{G}})$, and we write $\operatorname{Cov}(\mathbf{g}) = \mathbf{\Sigma}/B$ for some per-sample covariance $\mathbf{\Sigma}$. Let $\mathbf{H} \triangleq \nabla^2 L(\mathbf{w})$ be the Hessian at the current iterate. For the spike subspace, take the SVD of the mean gradient $\bar{\mathbf{G}} = \sum_{i=1}^{r} \sigma_i \mathbf{u}_i \mathbf{v}_i^\top$, define $\mathbf{s}_i \triangleq \operatorname{vec}(\mathbf{u}_i \mathbf{v}_i^\top)$, and let $\mathbf{\Pi}_k \triangleq \sum_{i=1}^{k} \mathbf{s}_i \mathbf{s}_i^\top$ be the projector onto $\operatorname{span}\{\mathbf{s}_1, \ldots, \mathbf{s}_k\}$. We denote the spike-restricted covariance by $\mathbf{\Sigma}_s \triangleq \mathbf{\Pi}_k \mathbf{\Sigma} \mathbf{\Pi}_k$. For simplicity, we first present the learning-rate bound under the minibatch-SGD update, which makes the role of spike-dominated gradient

variance explicit. A second-order expansion of $L$ around $\mathbf{w}$ and taking expectation yields a quadratic surrogate in $\eta$, whose minimizer gives the mean-optimal learning rate. The full proof is provided in Appendix A.4. We further provide an optimizer-agnostic extension in Appendix A.5.

**Theorem 2.1** (Spike-dominated variance bounds the smoothness-optimal learning rate). *Assume that the objective $L(\mathbf{w})$ is $\beta$-smooth. Let $\mathbf{g}$ be the mini-batch gradient with $\mathbb{E}[\mathbf{g}] = \bar{\mathbf{g}} = \nabla L(\mathbf{w})$ and $\operatorname{Cov}(\mathbf{g}) = \mathbf{\Sigma}/B$. Consider the SGD update $\mathbf{w}^+ = \mathbf{w} - \eta\mathbf{g}$. Then*

$$\mathbb{E}[L(\mathbf{w}^+)] \le L(\mathbf{w}) - \eta\|\bar{\mathbf{g}}\|_2^2 + \frac{\beta}{2}\eta^2\left(\|\bar{\mathbf{g}}\|_2^2 + \frac{1}{B}\operatorname{tr}(\mathbf{\Sigma})\right). \tag{1}$$

*The minimizer of this smoothness upper bound is*

$$\eta_{\mathrm{sm}}^* = \frac{\|\bar{\mathbf{g}}\|_2^2}{\beta\left(\|\bar{\mathbf{g}}\|_2^2 + \frac{1}{B}\operatorname{tr}(\mathbf{\Sigma})\right)}. \tag{2}$$

*Moreover, for any spike projector $\mathbf{\Pi}_k = \sum_{i=1}^{k} \mathbf{s}_i \mathbf{s}_i^\top$ and $\mathbf{\Sigma}_s = \mathbf{\Pi}_k \mathbf{\Sigma} \mathbf{\Pi}_k$, we have*

$$\eta_{\mathrm{sm}}^* \le \frac{\|\bar{\mathbf{g}}\|_2^2}{\beta\left(\|\bar{\mathbf{g}}\|_2^2 + \frac{1}{B}\operatorname{tr}(\mathbf{\Sigma}_s)\right)} \le \frac{B\|\bar{\mathbf{g}}\|_2^2}{\beta\operatorname{tr}(\mathbf{\Sigma}_s)} = \frac{B\|\bar{\mathbf{g}}\|_2^2}{\beta\sum_{i=1}^{k}\mathbf{s}_i^\top\mathbf{\Sigma}\mathbf{s}_i}. \tag{3}$$

*In addition, any learning rate satisfying*

$$0 < \eta < \frac{2\|\bar{\mathbf{g}}\|_2^2}{\beta\left(\|\bar{\mathbf{g}}\|_2^2 + \frac{1}{B}\operatorname{tr}(\mathbf{\Sigma})\right)} \tag{4}$$

*guarantees descent in the smoothness upper bound.*

Theorem 2.1 shows that the mean-optimal learning rate is governed by the curvature–variance denominator $\bar{\mathbf{g}}^\top\mathbf{H}\bar{\mathbf{g}} + \operatorname{tr}(\mathbf{\Sigma}\mathbf{H})/B$. When stochastic variance concentrates in the spike subspace, the effective bound tightens to the spike term $\operatorname{tr}(\mathbf{\Sigma}_s\mathbf{H})/B$, so $\eta^*$ is primarily constrained by common-structure fluctuations.

Taken together, *spike dominance suppresses long-tail learning through both local and global mechanisms: it contracts tail updates via spike-driven second-moment normalization, and it caps the globally stable step size through spike-dominated stochastic variance, leaving tail directions to evolve under persistently underpowered updates.*

## 2.4. Smaller Singular Directions Encode Sparser Semantics with Higher Relative Variance

This subsection examines tail singular directions as carriers of sparse semantic signals, where only a small fraction of samples yield non-negligible projections onto a given direction. We assess their reliability under stochastic gradients using *relative variance*, a scale-normalized measure of per-direction fluctuation.

We form a reference gradient $\bar{\mathbf{G}}$ by averaging gradients over a very large batch ($B_{\text{ref}} = 1024$) in Qwen3-0.6B, and compute its SVD $\bar{\mathbf{G}} = \sum_{k=1}^{r} \sigma_k \mathbf{u}_k \mathbf{v}_k^\top$ to define a fixed spectral basis. We then collect a large set of stochastic micro-batch gradients $\{\mathbf{G}_i\}$ and project each onto the $k$-th spectral component using $a_{i,k} \triangleq \mathbf{u}_k^\top \mathbf{G}_i \mathbf{v}_k$. We then compute the per-direction relative variance $\mathrm{RelVar}(k) \triangleq \frac{\mathrm{Var}(a_{i,k})}{\sigma_k^2}$, which measures the noise sensitivity of each spectral direction relative to its signal magnitude.

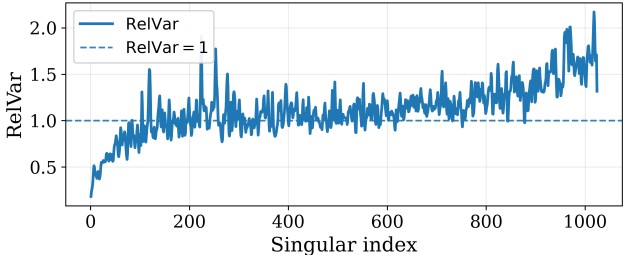

*Figure 4.* $\mathrm{RelVar}(k) = \mathrm{Var}(a_k)/\sigma_k^2$ increases with $k$, indicating more noise-dominated small-singular directions.

Figure 4 shows a monotonic rise of $\mathrm{RelVar}(k)$ with the singular-value index $k$ across micro-batch settings, with the increase most pronounced in the tail. Consequently, *smaller-singular directions are increasingly noise-dominated*: their stochastic fluctuations are large relative to their signal scale, consistent with tail components encoding sparser and more intermittent semantics.

### 2.5. Numerical Variance in Iterative Methods Disproportionately Rotates Tail Singular Directions

This subsection analyzes *numerical variance* introduced by iterative spectral routines. Using Newton–Schulz (NS) iteration as a representative example, we show that while spike directions remain relatively stable, tail directions can be substantially rotated by iterative updates.

Let $\mathrm{NS}(\mathbf{G})$ denote the matrix produced by NS iteration, and let $\{\mathbf{v}_i\}_{i=1}^{r}$ and $\{\widehat{\mathbf{v}}_i\}_{i=1}^{r}$ be the right singular vectors of $\mathbf{G}$ and $\mathrm{NS}(\mathbf{G})$, respectively. We quantify per-direction preservation by $\mathrm{align}(i) \triangleq \max_{j \in [r]} \left| \langle \mathbf{v}_i, \widehat{\mathbf{v}}_j \rangle \right|$, where a value close to 1 indicates the $i$-th direction is preserved, and a small value indicates severe rotation.

Figure 5 shows a clear head–tail split: leading singular vectors remain well aligned, around 0.85, while alignment drops monotonically in the tail and approaches 0.1.

Therefore, iterative spectral processing is not numerically neutral for long-tail learning: *it preferentially perturbs directions that already have weak and noise-sensitive signals, and more aggressive tail equalization would further exacerbate this effect while adding computation.*

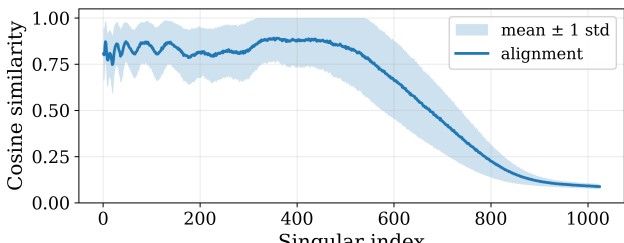

*Figure 5.* Alignment between singular directions of $G$ and $\mathrm{NS}(G)$. NS largely preserves head directions but severely disrupts tail directions.

## 3. Method

The findings in analysis section lead to a simple design principle: *attenuate the dominant common-feature spike subspace while avoiding aggressive amplification of the noise-dominated spectral tail.*

Motivated by this principle, we propose *Spectra*, a spike-aware optimizer that explicitly operates only on a low-dimensional spike subspace and leaves the tail unamplified. Spectra maintains a cached estimate of the spike subspace via warm-started power iteration, updates it intermittently, and performs spike singular-value shrinking towards the average scale of the tail.

### 3.1. The Spectra Optimizer

Spectra maintains a momentum matrix and performs spike singular-value shrinking on this momentum. At each step, it (i) updates the momentum, (ii) estimates a rank-$k$ spike subspace via warm-started power iteration, (iii) replaces the spike singular values with a tail-scale estimate while keeping the tail residual unchanged, and (iv) normalizes the step size using the RMS scale of the resulting shaped update. This suppresses spike-dominated updates without equalizing or amplifying the noise-sensitive tail, and avoids dense per-parameter second-moment statistics.

Unlike full spectrum-flattening, e.g. Muon, which enforces global equalization and can over-emphasize small-singular, noise-dominated modes, Spectra performs a *localized* intervention: it keeps the residual tail $\mathbf{M}_{\text{tail}}$ unchanged and only shrinks the spike singular values toward the tail's average scale $\sigma_{\text{tail}}$. The final update $\mathbf{O}_t$ is then used both for the parameter update and for RMS normalization, ensuring that step-size calibration reflects the shaped update actually applied.

### 3.2. Efficient Spike Subspace Estimation

A practical obstacle for spectral-domain optimization is the cost of repeatedly computing SVDs on large matrices. For $\mathbf{G} \in \mathbb{R}^{m \times n}$, a full SVD scales as $\mathcal{O}(\min(m,n)\,mn)$ and is infeasible in the training inner loop. However, gradient

**Algorithm 1** Spectra Optimizer Step

1: **Input:** weights $\mathbf{W}_{t-1} \in \mathbb{R}^{m \times n}$, gradient $\mathbf{G}_t \in \mathbb{R}^{m \times n}$, momentum $\mathbf{M}_{t-1} \in \mathbb{R}^{m \times n}$
2: **Hyperparams:** learning rate $\eta$, momentum coefficient $\mu$, rank ratio $r$, power iterations $T$, $\epsilon$
3: **Output:** updated weights $\mathbf{W}_t$, updated momentum $\mathbf{M}_t$
4: $k \leftarrow \max(1, \text{round}(r \cdot \min(m, n)))$
5: $\mathbf{M}_t \leftarrow \mu \mathbf{M}_{t-1} + \mathbf{G}_t$
6: $(\mathbf{U}_k, \mathbf{s}_k, \mathbf{V}_k) \leftarrow \text{POWERITERATIONSVD}(\mathbf{M}_t, k, T)$
7: $\mathbf{M}_{\text{tail}} \leftarrow \mathbf{M}_t - \mathbf{U}_k \, \text{diag}(\mathbf{s}_k) \, \mathbf{V}_k^\top$
8: $\sigma_{\text{tail}} \leftarrow \sqrt{\|\mathbf{M}_{\text{tail}}\|_F^2 / (\min(m, n) - k)}$
9: $\mathbf{O}_t \leftarrow \mathbf{M}_{\text{tail}} + \mathbf{U}_k \, \text{diag}(\sigma_{\text{tail}}\mathbb{I}) \, \mathbf{V}_k^\top$
10: $RMS \leftarrow \|\mathbf{O}_t\|_F / \sqrt{mn}$
11: $\eta' \leftarrow 0.2\,\eta / (RMS + \epsilon)$
12: $\mathbf{W}_t \leftarrow \mathbf{W}_{t-1} - \eta' \mathbf{O}_t$
13: **return** $(\mathbf{W}_t, \mathbf{M}_t)$

**Algorithm 2** Cached Power-Iteration SVD (rank-$k$)

1: **Input:** matrix $\mathbf{G} \in \mathbb{R}^{m \times n}$, rank $k$, iteration count $T$, cache state
2: **Output:** $(\mathbf{U}_k, \mathbf{s}_k, \mathbf{V}_k)$
3: $\mathbf{V}^{(0)} \leftarrow \text{State}[V_{\text{cache}}]$
4: **if** $\mathbf{V}^{(0)}$ is **None then**
5: $\quad (\mathbf{U}_k, \mathbf{s}_k, \mathbf{V}_k) \leftarrow \text{SVD\_LOWRANK}(\mathbf{G}, k)$ *// bootstrap*
6: $\quad \text{State}[V_{\text{cache}}] \leftarrow \mathbf{V}_k;$ **return** $(\mathbf{U}_k, \mathbf{s}_k, \mathbf{V}_k)$
7: **end if**
8: **for** $i = 1$ **to** $T$ **do**
9: $\quad \mathbf{P} \leftarrow \mathbf{G}\mathbf{V}^{(i-1)}$
10: $\quad \mathbf{U}^{(i)} \leftarrow \text{THINQR}(\mathbf{P})$
11: $\quad \mathbf{W} \leftarrow \mathbf{G}^\top \mathbf{U}^{(i)}$
12: $\quad \mathbf{s}^{(i)} \leftarrow \text{COLNORMS}(\mathbf{W})$
13: $\quad \mathbf{V}^{(i)} \leftarrow \mathbf{W} \, \text{diag}((\mathbf{s}^{(i)})^{-1})$ *// normalize columns*
14: **end for**
15: $\mathbf{U}_k \leftarrow \mathbf{U}^{(T)}, \mathbf{s}_k \leftarrow \mathbf{s}^{(T)}, \mathbf{V}_k \leftarrow \mathbf{V}^{(T)}$
16: $\text{State}[V_{\text{cache}}] \leftarrow \mathbf{V}_k$ *// update cache*
17: **return** $(\mathbf{U}_k, \mathbf{s}_k, \mathbf{V}_k)$

energy concentrates in a compact low-rank spike. Spectra therefore only needs a *rank-$k$* approximation that tracks this dominant subspace.

**Cached subspace iteration.** We estimate the spike subspace using a cached power-iteration routine (Algorithm 2). The key is to exploit temporal continuity: the spike subspace changes slowly across steps, so the previous right-singular subspace provides an accurate warm start. Concretely, we cache the rank-$k$ right subspace $\mathbf{V}$ from the previous step and initialize the current iteration with it. Each iteration applies two inexpensive projections, $\mathbf{P} \leftarrow \mathbf{G}\mathbf{V}$ and $\mathbf{W} \leftarrow \mathbf{G}^\top \mathbf{U}$, interleaved with a thin QR orthonormalization to maintain numerical stability. After $T$ iterations, we output a rank-$k$ estimate $(\mathbf{U}_k, \mathbf{s}_k, \mathbf{V}_k)$ and refresh the cache with $\mathbf{V}_k$. In practice, warm-starting substantially reduces the iterations required to reliably track the spike subspace compared to cold-start randomized SVD.

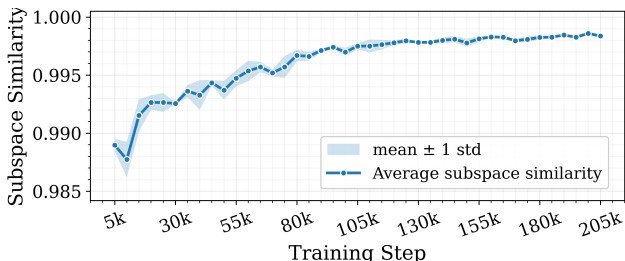

*Figure 6.* Temporal continuity of the spike subspace. We report the step-to-step similarity between the top-$k$ right-singular subspaces of consecutive gradients, showing consistently high similarity.

**Empirical justification: subspace continuity.** Caching is effective only if the spike subspace is stable across adjacent steps. We therefore measure the similarity between the spike right-singular subspaces of consecutive gradients using the largest canonical correlation. Figure 6 shows consistently

high similarity (above 0.98) throughout training, indicating that the spike subspace evolves slowly. This stability justifies cached warm-starts and enables accurate tracking with only one power-iteration step, amortizing the cost of subspace estimation in Spectra.

### 3.3. Efficiency Analysis

**Theoretical complexity.** Table 1 compares Spectra with AdamW (Loshchilov & Hutter, 2017) and Muon. AdamW applies element-wise moment updates and thus costs $\mathcal{O}(mn)$ per step, but stores two dense moment buffers ($2mn$). Muon performs Newton–Schulz iterations with full matrix multiplications, incurring $\mathcal{O}(T \cdot mn \min(m, n))$ time for $T$ iterations. Spectra estimates only a rank-$k$ spike subspace via cached power iteration: each iteration is dominated by two matrix multiplications $\mathbf{G}\mathbf{V}$ and $\mathbf{G}^\top \mathbf{U}$, costing $\mathcal{O}(mnk)$, with an additional $\mathcal{O}(mk^2)$ thin-QR negligible cost when $k$ is a small fraction of the layer dimension. Thus, Spectra's overhead is $\mathcal{O}(Tmnk)$ for $T$ power-iteration steps, with $k \approx 0.015 \min(m, n)$ in our default setting.

*Table 1.* Optimizer states memory cost and per-step complexity for $\mathbf{G} \in \mathbb{R}^{m \times n}$ ($k \ll \min(m, n)$). $T$ denote the iteration counts of NS (Muon) and power iteration (Spectra), respectively.

| Optimizer | Memory Cost | Complexity |
|-----------|-------------|------------|
| AdamW | $2mn$ | $\mathcal{O}(mn)$ |
| Muon | $mn$ | $\mathcal{O}(Tmn \min(m, n))$ |
| Spectra | $mn + nk$ | $\mathcal{O}(Tmnk)$ |

**Empirical Latency.** To validate the efficiency gains, we benchmark the latency of NS iterations against our Spectral Power Iteration on an NVIDIA H200 GPU. As shown in

Table 2, for a standard LLM layer size (4096 × 14336) in 8B model, Power-Iter with 1-2 iterations is 3.5× to 5× faster than NS. Even with 4 iterations, Spectra maintains a significant speed advantage. This efficiency allows Spectra to perform spectral preconditioning with negligible overhead compared to the backward pass.

# 4. Experiments

**Models and Datasets.** We conduct experiments on two architectures: Qwen3-0.6B on 100B tokens and LLaMA3-8B on 50B tokens. For pretraining, we use the DCLM (Li et al., 2024) dataset. For downstream evaluation, we consider three task types: question answering (ARC (Clark et al., 2018), RACE (Lai et al., 2017), BoolQ (Clark et al., 2019)), classification (HellaSwag (Zellers et al., 2019), PIQA (Bisk et al., 2020)), and cloze prediction (LAMBADA (Kazemi et al., 2023)). See Appendix A.6 for settings.

**Baselines.** We compare Spectra against AdamW and Muon, which applies iterative Newton–Schulz updates to approximate orthogonalized matrix steps; for Qwen3-0.6B, we additionally include Dion (Ahn et al., 2025), which uses power iteration to form a low-rank update.

## 4.1. Main Results

**Training Loss and Convergence.** Figure 7 illustrates the training loss curves for both the 0.6B and 8B models. Spectra demonstrates superior convergence efficiency across both scales: on the 0.6B model, Spectra achieves a final validation loss that is 2.1% lower than AdamW and 1.4% lower than Muon, and reaches a matched loss level 30% faster in wall-clock time than AdamW. This scaling advantage is further confirmed on the 8B model, where Spectra outperforms AdamW and Muon by 1.5% and 0.4% in final loss.

**Downstream Performance.** Table 3 summarizes the performance across all benchmarks. Spectra consistently outperforms both AdamW and Muon at both scales. On Qwen3-0.6B, Spectra improves average accuracy by +1.41% over AdamW and +0.89% over Muon. On LLaMA3-8B, Spectra also achieves the best average accuracy, improving over AdamW and Muon by +1.62% and +0.66%, respectively. Together, these gains persist as we scale from 0.6B to 8B.

**Wall-Clock Speedup.** Beyond the 0.6B and LLaMA3-8B settings in Figure 7, we further evaluate time-to-target convergence on Qwen3-2B-A0.8B and Qwen3-8B. As shown in Table 5, Spectra consistently reaches the same loss levels faster than AdamW and Muon. On Qwen3-0.6B, Spectra achieves up to 1.31× speedup over AdamW; on Qwen3-2B-A0.8B and Qwen3-8B, it achieves up to 1.34× and 1.24× speedup, respectively. Its advantage over Muon also increases with scale. These results indicate that Spectra is not only more efficient per step, but also reduces end-to-end

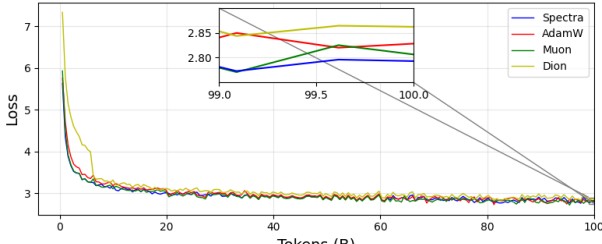

(A) Qwen3-0.6B

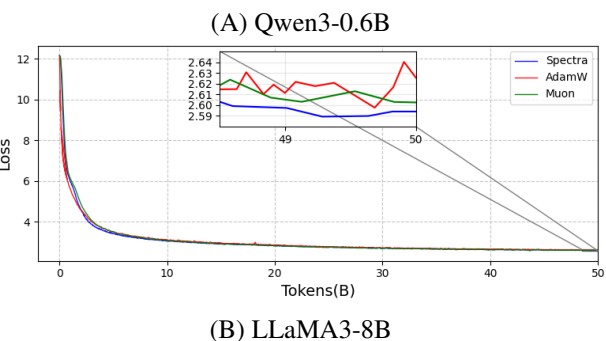

(B) LLaMA3-8B

*Figure 7.* Training loss curves for (A) Qwen3-0.6B on 100B tokens and (B) LLaMA3-8B on 50B tokens.

pretraining cost through faster wall-clock convergence.

**Computational Efficiency.** We further compare the end-to-end per-step runtime of Spectra with AdamW and Muon across Qwen3 models from 0.6B to 32B using NVIDIA H200 GPUs. As shown in Table 4, Spectra consistently achieves the lowest per-step runtime among the three optimizers. This indicates that Spectra is not only faster than Muon in optimizer processing, but also more lightweight than AdamW in end-to-end training.

## 4.2. Ablation Study

To further quantify the factors contributing to the effectiveness of Spectra, we conduct a series of ablation studies on the Qwen3-0.6B model trained with 50B tokens. We investigate the impact of the rank ratio for head compression, the number of power iterations, and the optimizer's sensitivity to learning rates. The results across downstream tasks are summarized in Table 6 and Figure 8. In addition, we provide a separate ablation on spike-value smoothing strategies under a 20B-token setting in Appendix A.7.

**Sensitivity to Rank Ratio ($r$).** We vary the rank ratio $r \in \{1.5\%, 5\%, 10\%, 15\%\}$ used for spike compression. As shown in Table 6 and Figure 8, downstream performance is largely insensitive to $r$: the average accuracy changes by less than 0.25 points across the full range, and individual tasks exhibit only small fluctuations. This suggests that even a minimal rank ratio ($r = 1.5\%$) already captures the dominant spike directions, and increasing $r$ provides limited practical benefit. We therefore use $r = 1.5\%$ by default to

*Table 2.* Empirical latency comparison (ms) on H200 GPU. Power-Iter uses a rank ratio of $1.5\%$. $T$ denotes the number of power iterations.

| Matrix Size | NS | Power-Iter ($T=1$) | Power-Iter ($T=2$) | Power-Iter ($T=4$) | Power-Iter ($T=8$) |
|---|---|---|---|---|---|
| (4096, 4096) | 3.5664 | 0.9724 | 1.6012 | 2.8648 | 5.3927 |
| (4096, 14336) | 9.1465 | 1.7799 | 2.5622 | 4.1315 | 7.2676 |

*Table 3.* Downstream performance comparison for Qwen3-0.6B trained on 100B and LLaMA3-8B trained on 50B tokens. Spectra consistently achieves the highest average accuracy across both model scales.

| Model | Optimizer | Loss | ArcC | ArcE | BoolQ | HellaSwag | LAMBADA | PIQA | RACE | Avg |
|---|---|---|---|---|---|---|---|---|---|---|
| Qwen3-0.6B | AdamW | 2.83 | **29.95** | 54.34 | 57.00 | 51.21 | 46.87 | 72.09 | 49.41 | 51.55 |
| | Dion | 2.86 | 27.82 | 53.41 | 53.12 | 49.25 | 44.75 | 70.84 | 49.41 | 49.80 |
| | Muon | 2.81 | 29.61 | 54.50 | 56.91 | **52.80** | **48.81** | 72.09 | 49.80 | 52.07 |
| | Spectra | **2.77** | 29.61 | **56.61** | **60.21** | 52.30 | 48.44 | **73.18** | **50.34** | **52.96** |
| LLaMA3-8B | AdamW | 2.63 | 38.40 | 65.11 | 57.23 | 61.29 | 48.27 | 75.21 | 49.21 | 56.39 |
| | Muon | 2.60 | **38.77** | 65.94 | 59.84 | 61.43 | 47.97 | 75.13 | **52.36** | 57.35 |
| | Spectra | **2.59** | 38.46 | **67.11** | **60.08** | **62.95** | **49.17** | **76.43** | 51.88 | **58.01** |

*Table 4.* End-to-end per-step runtime comparison across Qwen3 models. Runtime is measured in milliseconds per training step. Runtime reduction is computed relative to Spectra.

| Model | AdamW | Muon | Spectra | Runtime Reduction | |
|---|---|---|---|---|---|
| | Per-Step Runtime (ms) | | | vs. AdamW | vs. Muon |
| Qwen3-0.6B | 5297.0 | 5385.0 | **5259.0** | 0.72% | 2.34% |
| Qwen3-2B-A0.8B | 4312.9 | 6064.7 | **3973.7** | 7.86% | 34.48% |
| Qwen3-8B | 10143.0 | 10243.0 | **9750.0** | 3.88% | 4.81% |
| Qwen3-32B | 16308.0 | 16690.5 | **16275.9** | 0.20% | 2.48% |

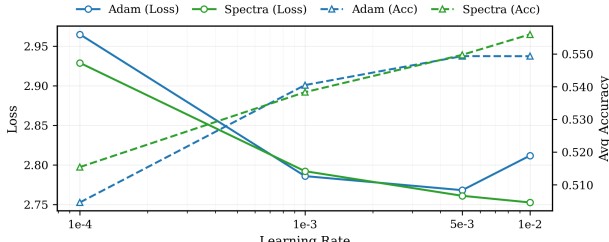

*Figure 8.* Comparison of Spectra and AdamW across learning rates $\eta \in \{8 \times 10^{-4}, 1 \times 10^{-3}, 5 \times 10^{-3}, 1 \times 10^{-2}\}$. Spectra shows superior convergence loss and downstream performance in most regimes.

minimize overhead.

**Number of Power Iterations ($T$).** We ablate the number of power-iteration steps $T \in \{1, 2, 4, 8\}$ used for subspace estimation. As shown in Table 6, increasing $T$ does not improve the overall downstream average and can slightly degrade performance, with the drop largely driven by BoolQ; other tasks exhibit only minor variations. Since $T = 1$ also has the lowest compute overhead, we adopt a single cached power-iteration step by default.

**Spike-Value Treatment.** We ablate several spike-value treatments, including replacing the spike with its minimum singular value, hard removal of the spike component, soft-clipping, and our tail-average replacement. As shown in Appendix A.7, two conclusions are clear. First, suppressing the spike scale is fairly robust: multiple rescaling variants outperform hard removal. Second, zeroing out the spike is clearly worse than rescaling it, confirming that spike directions still contain useful optimization signal and should not be discarded entirely.

**Learning Rate Robustness.** We sweep the learning rate $\eta \in \{8\times10^{-4}, 1\times10^{-3}, 5\times10^{-3}, 1\times10^{-2}\}$ and compare against AdamW. Figure 8 shows that Spectra remains stable across this wide range and achieves better loss/accuracy at most settings. At $\eta = 1 \times 10^{-2}$, where AdamW exhibits signs of instability, Spectra maintains robust convergence, indicating improved tolerance to larger step sizes.

## 5. Related Work

**Element-wise adaptive optimization.** Adaptive methods such as AdamW (Loshchilov & Hutter, 2017) rescale updates using per-parameter moment statistics and are widely used for LLM training. However, operating at the coordinate level, they ignore structured correlations within matrix-valued parameters. When gradient energy concentrates in a low-rank subspace, element-wise normalization becomes dominated by these directions, suppressing long-tail updates. Spectra differs by explicitly operating in the spectral domain and directly targeting this low-rank anisotropy.

**Matrix-aware preconditioning.** Matrix-structured opti-

*Table 5.* Time-to-target comparison across Qwen3 models. We report wall-clock hours required to reach each target loss. Speedup is computed as baseline time divided by Spectra time.

| Model | Target Loss | AdamW (h) | Muon (h) | Spectra (h) | vs. AdamW | vs. Muon |
|---|---|---|---|---|---|---|
| Qwen3-0.6B | 3.7 | 7.29 | 5.91 | **5.22** | **1.40×** | **1.13×** |
| | 3.5 | 10.73 | 7.85 | **7.65** | **1.40×** | **1.03×** |
| | 3.3 | 16.31 | 14.21 | **13.22** | **1.23×** | **1.07×** |
| | 3.0 | 73.57 | 58.26 | **56.01** | **1.31×** | **1.04×** |
| Qwen3-2B-A0.8B | 3.7 | 2.75 | 2.41 | **2.18** | **1.26×** | **1.11×** |
| | 3.5 | 4.21 | 3.55 | **3.39** | **1.24×** | **1.05×** |
| | 3.3 | 8.13 | 6.45 | **6.05** | **1.34×** | **1.07×** |
| Qwen3-8B | 3.7 | 9.58 | 10.52 | **8.01** | **1.20×** | **1.31×** |
| | 3.5 | 12.40 | 11.76 | **10.37** | **1.20×** | **1.13×** |
| | 3.3 | 17.83 | 16.08 | **13.99** | **1.27×** | **1.15×** |
| | 3.1 | 27.90 | 24.50 | **22.54** | **1.24×** | **1.09×** |

*Table 6.* Ablation studies on Qwen3-0.6B. We explore different rank ratios ($r$) and power iteration counts ($T$). All models are trained on 50B tokens.

| Ablation | Setting | ArcC | ArcE | BoolQ | HellaSwag | LAMBADA | PIQA | RACE | Avg |
|---|---|---|---|---|---|---|---|---|---|
| Rank Ratio $r$ | 1.5% (Default) | **28.92** | 51.52 | 58.93 | 48.52 | **46.34** | 71.33 | 49.49 | 50.72 |
| | 5% | 26.37 | 52.40 | **61.31** | 48.89 | 45.22 | 70.95 | 49.57 | 50.67 |
| | 10% | 28.24 | 53.07 | 59.60 | 49.12 | 45.72 | 70.73 | 49.33 | 50.83 |
| | 15% | 27.99 | **53.45** | 57.83 | **49.45** | 46.28 | **71.55** | **49.86** | **50.92** |
| Power Iter $T$ | $T = 1$ (Default) | **28.92** | 51.52 | **58.93** | 48.52 | **46.34** | **71.33** | 49.49 | **50.72** |
| | $T = 2$ | 27.73 | **52.27** | 56.15 | **48.96** | 45.20 | 70.35 | **50.18** | 50.12 |
| | $T = 4$ | 27.47 | 52.19 | 53.82 | 48.76 | 45.39 | 70.62 | 50.04 | 49.76 |
| | $T = 8$ | 28.41 | 52.15 | 50.52 | 47.75 | 45.74 | 69.59 | 50.00 | 49.17 |

mizers, including Shampoo (Gupta et al., 2018) and SOAP (Vyas et al., 2024), capture cross-coordinate correlations via Kronecker-factored or second-order statistics. While effective, their memory and computational costs scale poorly with layer dimensions, limiting practicality for large-scale LLM pretraining. Spectra avoids approximating full second-order structure by exploiting the empirical low-rank concentration of gradients, enabling efficient low-rank spectral shaping with minimal overhead.

**Orthogonal and spectral update methods.** Recent methods such as Muon (Jordan et al.), Dion (Ahn et al., 2025), and PolarGrad (Lau et al., 2025) reduce gradient anisotropy through orthogonalization or spectrum flattening. These approaches typically apply global spectral transformations, which can amplify noise-dominated small-singular directions and incur substantial numerical cost. In contrast, Spectra performs a localized intervention suppressing the dominant spike subspace without amplifying the spectral tail.

## 6. Conclusion

We identify a persistent spike–tail structure in LLM gradients, where a small low-rank subspace dominates optimization and suppresses learning in the long tail. To address this asymmetry, we propose Spectra, a spike-aware optimizer that selectively suppresses dominant spectral components without amplifying noise-sensitive tail directions. By reshaping the gradient spectrum in this targeted manner, Spectra yields more benign conditioning in practice, enabling larger stable learning rates, faster convergence, and improved downstream performance with minimal computational and memory overhead. These results suggest that viewing gradients as structured spectral objects, rather than independent coordinates, offers a principled basis for scalable and robust LLM optimization, and that selectively targeting dominant structure can improve both stability and efficiency in practice.

## Acknowledgements

This research is supported in part by the National Natural Science Foundation of China under Grant 62090025.

## Impact Statement

This paper presents work whose goal is to advance the field of machine learning. There are many potential societal consequences of our work, none of which we feel must be specifically highlighted here.

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

# A. Appendix.

## A.1. Gradient Anisotropy

To complement Figure 1 (deepest MLP), we report the gradient singular spectrum for three additional parameter matrices: (i) the shallowest attention $k$-projection, (ii) the shallowest MLP up-projection, and (iii) the deepest attention $k$-projection. Each plot overlays spectra at initialization and at convergence, and includes four Qwen3 model scales.

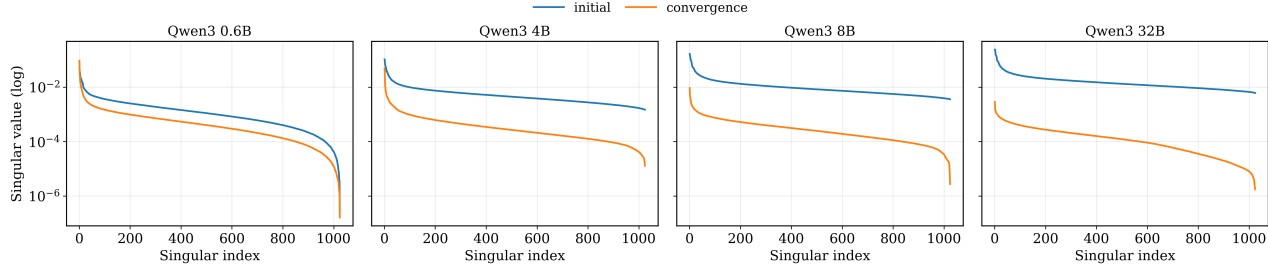

*Figure 9.* Gradient spectrum (initial vs. convergence) for the shallowest attention layer (self-attention $k$-projection) across four Qwen3 model scales.

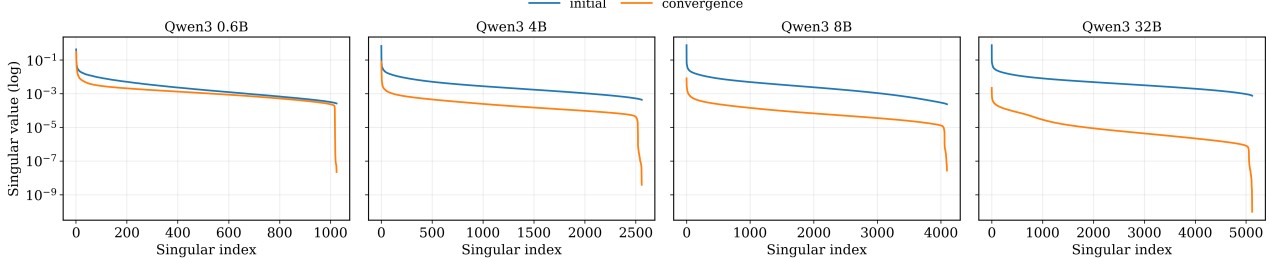

*Figure 10.* Gradient spectrum (initial vs. convergence) for the shallowest MLP layer (up-projection) across four Qwen3 model scales.

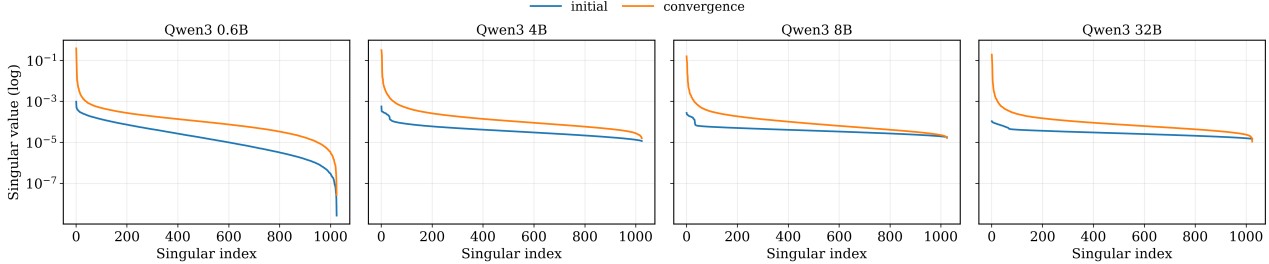

*Figure 11.* Gradient spectrum (initial vs. convergence) for a deeper attention layer (self-attention $k$-projection) across four Qwen3 model scales.

## A.2. Semantic Correspondence of Gradient Anisotropy

Figure 12 reports the controlled-intervention results on LLaMA3-8B, matching the description in Section 2. Frequency-normalized loss (*FreqNorm*) selectively suppresses the leading spike components, while intra-sentence token permutation (*Shuffle*) selectively amplifies them; in both cases, changes rapidly vanish in the tail.

## A.3. Spike Updating Suppresses Long-Tail Learning on LLaMA3-8B

Figure 13 reports the same analysis as Figure 3, but on LLaMA3-8B. We show (top) the cumulative spectral energy (CDF) of AdamW moments and (bottom) the distribution of tail updates under full normalization versus a tail-only baseline.

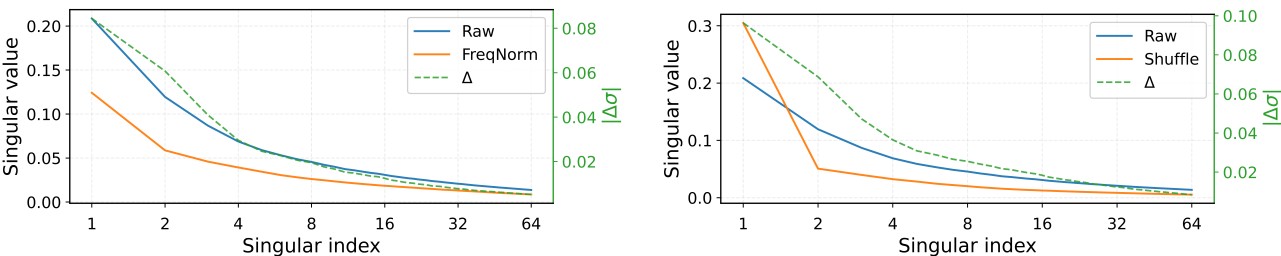

*Figure 12.* Gradient spectrum under two controlled interventions on LLaMA3-8B. *Left:* frequency-normalized loss (*FreqNorm*) selectively suppresses the leading spike components. *Right:* intra-sentence token permutation (*Shuffle*) selectively amplifies them; in both cases, changes rapidly vanish in the tail.

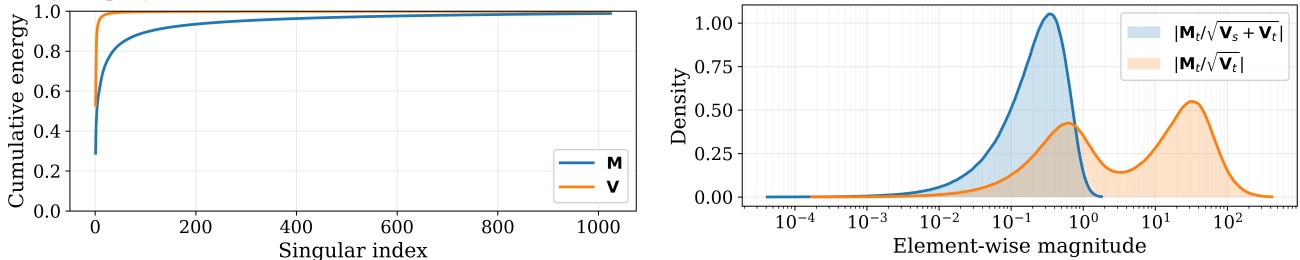

*Figure 13.* Spike-dominated second-moment accumulation suppresses tail updates (LLaMA3-8B). *Left:* cumulative spectral energy (CDF) of AdamW moments. *Right:* element-wise magnitudes of tail updates under full normalization versus the tail-only baseline.

## A.4. Proof of Theorem 2.1

*Proof of Theorem 2.1.* Let $\mathbf{g}$ denote the mini-batch gradient, with $\mathbb{E}[\mathbf{g}] = \bar{\mathbf{g}} = \nabla L(\mathbf{w})$ and $\mathrm{Cov}(\mathbf{g}) = \boldsymbol{\Sigma}/B$. Consider the SGD update $\mathbf{w}^+ = \mathbf{w} - \eta\mathbf{g}$.

**Step 1: smoothness upper bound.** Since $L$ is $\beta$-smooth, for any update direction $\mathbf{u}$,

$$L(\mathbf{w} + \mathbf{u}) \le L(\mathbf{w}) + \nabla L(\mathbf{w})^\top \mathbf{u} + \frac{\beta}{2}\|\mathbf{u}\|_2^2. \tag{5}$$

Taking $\mathbf{u} = -\eta\mathbf{g}$ gives

$$L(\mathbf{w} - \eta\mathbf{g}) \le L(\mathbf{w}) - \eta\,\nabla L(\mathbf{w})^\top \mathbf{g} + \frac{\beta}{2}\eta^2\|\mathbf{g}\|_2^2. \tag{6}$$

Taking expectation over the mini-batch randomness and using $\nabla L(\mathbf{w}) = \bar{\mathbf{g}}$ yields

$$\mathbb{E}[L(\mathbf{w} - \eta\mathbf{g})] \le L(\mathbf{w}) - \eta\,\bar{\mathbf{g}}^\top\mathbb{E}[\mathbf{g}] + \frac{\beta}{2}\eta^2\mathbb{E}[\|\mathbf{g}\|_2^2]$$

$$= L(\mathbf{w}) - \eta\|\bar{\mathbf{g}}\|_2^2 + \frac{\beta}{2}\eta^2\mathbb{E}[\|\mathbf{g}\|_2^2]. \tag{7}$$

Using the second-moment decomposition,

$$\mathbb{E}[\|\mathbf{g}\|_2^2] = \|\mathbb{E}[\mathbf{g}]\|_2^2 + \mathrm{tr}(\mathrm{Cov}(\mathbf{g})) = \|\bar{\mathbf{g}}\|_2^2 + \frac{1}{B}\mathrm{tr}(\boldsymbol{\Sigma}), \tag{8}$$

we obtain

$$\mathbb{E}[L(\mathbf{w}^+)] \le L(\mathbf{w}) - \eta\|\bar{\mathbf{g}}\|_2^2 + \frac{\beta}{2}\eta^2\left(\|\bar{\mathbf{g}}\|_2^2 + \frac{1}{B}\mathrm{tr}(\boldsymbol{\Sigma})\right), \tag{9}$$

which proves Eq. (1).

**Step 2: minimizing the smoothness upper bound.** The right-hand side of Eq. (1) is a quadratic function of $\eta$. Its minimizer is

$$\eta_{\mathrm{sm}}^* = \frac{\|\bar{\mathbf{g}}\|_2^2}{\beta\left(\|\bar{\mathbf{g}}\|_2^2 + \frac{1}{B}\mathrm{tr}(\boldsymbol{\Sigma})\right)}, \tag{10}$$

which proves Eq. (2). Moreover, the upper bound decreases relative to $L(\mathbf{w})$ whenever

$$-\eta\|\bar{\mathbf{g}}\|_2^2 + \frac{\beta}{2}\eta^2\left(\|\bar{\mathbf{g}}\|_2^2 + \frac{1}{B}\mathrm{tr}(\boldsymbol{\Sigma})\right) < 0. \tag{11}$$

Solving this inequality for $\eta > 0$ gives

$$0 < \eta < \frac{2\|\bar{\mathbf{g}}\|_2^2}{\beta\left(\|\bar{\mathbf{g}}\|_2^2 + \frac{1}{B}\mathrm{tr}(\boldsymbol{\Sigma})\right)}, \tag{12}$$

which proves Eq. (4).

**Step 3: spike-restricted variance bounds the learning rate.** Let $\boldsymbol{\Pi}_k = \sum_{i=1}^k \mathbf{s}_i\mathbf{s}_i^\top$ be the orthogonal projector onto the spike subspace, and define $\boldsymbol{\Sigma}_s = \boldsymbol{\Pi}_k\boldsymbol{\Sigma}\boldsymbol{\Pi}_k$. Since $\boldsymbol{\Sigma} \succeq 0$ and $0 \preceq \boldsymbol{\Pi}_k \preceq \mathbf{I}$,

$$\mathrm{tr}(\boldsymbol{\Sigma}_s) = \mathrm{tr}(\boldsymbol{\Pi}_k\boldsymbol{\Sigma}\boldsymbol{\Pi}_k) = \mathrm{tr}(\boldsymbol{\Pi}_k\boldsymbol{\Sigma}) \le \mathrm{tr}(\boldsymbol{\Sigma}). \tag{13}$$

Therefore,

$$\eta_{\mathrm{sm}}^* = \frac{\|\bar{\mathbf{g}}\|_2^2}{\beta\left(\|\bar{\mathbf{g}}\|_2^2 + \frac{1}{B}\mathrm{tr}(\boldsymbol{\Sigma})\right)} \le \frac{\|\bar{\mathbf{g}}\|_2^2}{\beta\left(\|\bar{\mathbf{g}}\|_2^2 + \frac{1}{B}\mathrm{tr}(\boldsymbol{\Sigma}_s)\right)}. \tag{14}$$

Dropping the nonnegative term $\|\bar{\mathbf{g}}\|_2^2$ in the denominator gives

$$\eta_{\mathrm{sm}}^* \le \frac{B\|\bar{\mathbf{g}}\|_2^2}{\beta\,\mathrm{tr}(\boldsymbol{\Sigma}_s)}. \tag{15}$$

Finally, using $\boldsymbol{\Sigma}_s = \boldsymbol{\Pi}_k\boldsymbol{\Sigma}\boldsymbol{\Pi}_k$ and $\boldsymbol{\Pi}_k = \sum_{i=1}^k \mathbf{s}_i\mathbf{s}_i^\top$ with orthonormal $\{\mathbf{s}_i\}_{i=1}^k$, we have

$$\mathrm{tr}(\boldsymbol{\Sigma}_s) = \mathrm{tr}(\boldsymbol{\Pi}_k\boldsymbol{\Sigma}) = \sum_{i=1}^k \mathbf{s}_i^\top\boldsymbol{\Sigma}\mathbf{s}_i. \tag{16}$$

Substituting this identity proves Eq. (3) and completes the proof. $\square$

### A.5. Optimizer-Agnostic Learning-Rate Bound

We provide a general optimizer-agnostic version of the second-order learning-rate argument. The main theorem in Section 2 is stated for the minibatch-SGD update for simplicity, since this form makes the contribution of spike-dominated gradient variance explicit. However, the same reasoning applies to any stochastic optimizer update direction.

Let $L(\mathbf{w})$ be the objective, and denote

$$\mathbf{g} = \nabla L(\mathbf{w}), \qquad \mathbf{H} = \nabla^2 L(\mathbf{w}).$$

Consider a one-step update of the form

$$\mathbf{w}^+ = \mathbf{w} - \eta\boldsymbol{\phi},$$

where $\boldsymbol{\phi}$ is the stochastic update direction produced by an arbitrary optimizer. Define

$$\mathbf{m}_\phi = \mathbb{E}[\boldsymbol{\phi}], \qquad \mathbf{Q}_\phi = \mathbb{E}[\boldsymbol{\phi}\boldsymbol{\phi}^\top].$$

A second-order expansion gives

$$L(\mathbf{w} - \eta\boldsymbol{\phi}) \approx L(\mathbf{w}) - \eta\,\mathbf{g}^\top\boldsymbol{\phi} + \frac{1}{2}\eta^2\boldsymbol{\phi}^\top\mathbf{H}\boldsymbol{\phi}. \tag{17}$$

Taking expectation over the stochasticity of $\boldsymbol{\phi}$ yields

$$\mathbb{E}[L(\mathbf{w} - \eta\boldsymbol{\phi})] \approx L(\mathbf{w}) - \eta\,\mathbf{g}^\top\mathbf{m}_\phi + \frac{1}{2}\eta^2\mathrm{tr}(\mathbf{H}\mathbf{Q}_\phi). \tag{18}$$

When $\mathrm{tr}(\mathbf{H}\mathbf{Q}_\phi) > 0$, the minimizer of this quadratic surrogate is

$$\eta_\phi^\star = \frac{\mathbf{g}^\top\mathbf{m}_\phi}{\mathrm{tr}(\mathbf{H}\mathbf{Q}_\phi)}. \tag{19}$$

This expression shows that the mean-optimal learning rate is controlled by the alignment term $\mathbf{g}^\top \mathbf{m}_\phi$ in the numerator and the curvature-weighted second moment of the update direction in the denominator.

Theorem 2.1 is recovered as the minibatch-SGD special case. Let $\phi = \mathbf{g}_B$ be the minibatch gradient, with

$$\mathbb{E}[\mathbf{g}_B] = \bar{\mathbf{g}}, \qquad \mathbb{E}[\mathbf{g}_B \mathbf{g}_B^\top] = \bar{\mathbf{g}}\bar{\mathbf{g}}^\top + \frac{1}{B}\mathbf{\Sigma}.$$

Substituting these into Eq. (19) gives

$$\eta^\star = \frac{\|\bar{\mathbf{g}}\|_2^2}{\bar{\mathbf{g}}^\top \mathbf{H}\bar{\mathbf{g}} + \frac{1}{B}\mathrm{tr}(\mathbf{\Sigma}\mathbf{H})}, \tag{20}$$

which matches Eq. (??). Therefore, the SGD theorem is a concrete instance of the more general update-direction result.

For AdamW, the stochastic update direction can be instantiated by the actual preconditioned update. Ignoring the deterministic decoupled weight-decay term, AdamW uses

$$\mathbf{m}_t = \beta_1 \mathbf{m}_{t-1} + (1 - \beta_1)\mathbf{g}_t, \qquad \mathbf{v}_t = \beta_2 \mathbf{v}_{t-1} + (1 - \beta_2)(\mathbf{g}_t \odot \mathbf{g}_t),$$

with bias-corrected moments

$$\widehat{\mathbf{m}}_t = \frac{\mathbf{m}_t}{1 - \beta_1^t}, \qquad \widehat{\mathbf{v}}_t = \frac{\mathbf{v}_t}{1 - \beta_2^t}.$$

The corresponding stochastic update direction is

$$\phi_t = \frac{\widehat{\mathbf{m}}_t}{\sqrt{\widehat{\mathbf{v}}_t} + \epsilon}. \tag{21}$$

Thus Eq. (19) applies with

$$\mathbf{m}_\phi = \mathbb{E}[\phi_t], \qquad \mathbf{Q}_\phi = \mathbb{E}[\phi_t \phi_t^\top].$$

Consequently, AdamW also admits the same form of mean-optimal learning rate,

$$\eta_\phi^\star = \frac{\mathbf{g}^\top \mathbb{E}[\phi_t]}{\mathrm{tr}\left(\mathbf{H}\,\mathbb{E}[\phi_t \phi_t^\top]\right)}. \tag{22}$$

Therefore, if the actual AdamW update second moment or covariance is spike-dominant, then the same curvature-weighted second-moment term constrains the stable and effective learning-rate range.

## A.6. Experiment details

The detailed training configurations of Qwen3-0.6B and LLaMA3-8B are shown in Table 7 and Table 8.

*Table 7.* Model configurations for Qwen3-0.6B.

| Configurations | Qwen3-0.6B |
|---|---|
| Hidden activation | silu |
| Max position embeddings | 40960 |
| Vocabulary size | 151936 |
| Sequence length | 1024 |
| LayerNorm $\epsilon$ (rms_norm_eps) | $1 \times 10^{-6}$ |
| Dropout probability (attention_dropout) | 0.0 |
| LR warm-up steps | 2,000 |
| LR scheduler | Cosine |
| Hidden size | 1024 |
| Intermediate size | 3072 |
| Hidden layers | 28 |
| Num. attention heads | 16 |
| Batch Size | 512 |

*Table 8.* Model configurations for LLaMA3-8B.

| Configurations | LLaMA3-8B |
|---|---|
| Hidden activation | silu |
| Max position embeddings | 8192 |
| Vocabulary size | 128256 |
| Sequence length | 4,096 |
| LayerNorm $\epsilon$ (rms_norm_eps) | $1 \times 10^{-5}$ |
| Dropout probability (attention_dropout) | 0.0 |
| LR warm-up steps | 7629 |
| LR scheduler | Cosine |
| Hidden size | 4096 |
| Intermediate size | 14336 |
| Hidden layers | 32 |
| Num. attention heads | 32 |
| Batch Size | 512 |

*Table 9.* Downstream performance under different spike-value treatments on Qwen3-0.6B trained with 20B tokens. The results show that rescaling the spike is preferable to removing it entirely.

| Spike Treatment | Loss | ArcC | ArcE | HellaSwag | LAMBADA | PIQA | RACE | Avg |
|---|---|---|---|---|---|---|---|---|
| Spike-min | 3.06 | 26.62 | 49.07 | 43.27 | 40.66 | 68.93 | 49.41 | 46.33 |
| Zero-spike | 3.07 | 24.32 | 46.17 | 37.98 | 34.43 | 66.92 | 49.15 | 43.16 |
| Soft-clipping | **3.00** | 25.26 | 49.16 | **43.38** | **41.65** | **69.64** | **49.47** | 46.43 |
| Ours | 3.01 | **26.79** | **49.92** | 42.84 | 41.53 | 69.04 | 49.19 | **46.55** |

## A.7. Ablation on Spike-Value Treatment

We further study how the identified spike component should be treated once its subspace has been estimated. This ablation is conducted on Qwen3-0.6B trained with 20B tokens, and is therefore reported separately from the 50B-token ablations in the main text. The purpose of this experiment is to isolate the effect of different smoothing strategies under the same training budget, rather than to directly compare absolute downstream scores with the 50B-token setting.

The spike is not always harmful. On the contrary, it contains important common gradient structure. The issue is not the existence of the spike itself, but its excessively large scale. Therefore, the goal is not to remove the spike, but to rescale it to a more appropriate level. We compare four treatments: *spike-min*, which replaces the spike singular values with the minimum singular value within the spike subspace; *zero-spike*, which removes the spike component entirely; *soft-clipping*, which applies a continuous shrinkage scheme; and *ours*, which replaces the spike singular values with the average scale of the spectral tail.

As shown in Table 9, two conclusions are clear. First, suppressing the spike scale is fairly robust: several rescaling-based variants achieve similar downstream performance and substantially outperform hard removal. Second, zeroing the spike is clearly worse than rescaling it, indicating that spike directions still contain useful signal and should not be discarded entirely. Among the compared variants, our tail-average replacement achieves the best average downstream accuracy.

