# OpenReview forum: "Spectra: Rethinking Optimizers for LLMs Under Spectral Anisotropy"
_ICML.cc/2026/Conference — ICML 2026 regular_

### Official Review · Reviewer_Fwmz · 2026-03-06

**Soundness:** 3
**Presentation:** 3
**Significance:** 3
**Originality:** 2
**Overall Recommendation:** 5
**Confidence:** 4

**Summary:**

The paper explores spectral anisotropy in LLM training gradients, identifying a "spike-tail" structure where a small fraction of singular directions dominate optimizer statistics. They propose Spectra, an optimizer that shrinks spike singular values toward the tail RMS scale. The paper reports results on Qwen3-0.6B and LLaMA3-8B pretraining, showing improved computational efficiency and training loss compared to Muon and AdamW.

**Compliance With Llm Reviewing Policy:**

Affirmed.

**Key Questions For Authors:**

see Strengths And Weaknesses, 2 questions for code and hyperparameter

**Limitations:**

yes

**Strengths And Weaknesses:**

**Strengths:**
The paper is clearly written. The topic is interesting: reshaping the gradient spectrum for better optimization dynamics. The ablation studies are thorough.

**Weaknesses:**
1. No major weaknesses, only a couple of questions.
2. I carefully examined the Megatron code and found a subtle implementation problem. If I understand correctly, the Nesterov momentum is implemented differently for Muon and Spectra. For Muon, it uses `exp_avg.lerp_(grad, 1 - group["momentum_beta"])`, while for Spectra it uses `buf.mul_(momentum).add_(g)`. These correspond to $(1-\beta) \cdot G_t + \beta \cdot M_{t-1}$ versus $G_t + \mu \cdot M_{t-1}$, which are not equivalent. For $\beta = \mu = 0.95$, Spectra assigns roughly 20× more weight to the current gradient $G_t$ than Muon does.
3. Spectral reshaping is intuitive, but Spectra's effective update RMS scale per matrix may differ from Muon's. I am curious whether the experiments in Section 4.1 used matched learning rates across AdamW, Muon, Dion, and Spectra, or whether update RMS was equalized as a fairer comparison baseline, to avoid confounds from hyperparameter differences.

---

> ### Author Rebuttal · Authors · 2026-03-30
>
> Thank you very much for your careful review. We summarize our responses as follows:
>
> **Different momentum forms in Muon vs. Spectra:** The two recursions differ only by a constant scale factor; they remain collinear at every step and therefore are equivalent in direction and relative weighting.
>
> **Concern of hyperparameter differences:** We match the effective update RMS under the same learning-rate schedule.
>
> *Q1: The Nesterov momentum is implemented differently for Muon and Spectra.*
>
> **A1:** The key point is that these two momentum implementations are equivalent up to a constant scalar factor, not different in direction or relative weighting.
>
> The two implementations are indeed written in different algebraic forms:
>
> **Muon:** $m_t=\beta m_{t-1}+(1-\beta)g_t$
>
> **Spectra:** $b_t=\beta b_{t-1}+g_t$
>
> Starting from zero initialization, unrolling the recursion gives
>
> $m_t=(1-\beta)\sum_{i=1}^{t}\beta^{\,t-i} g_i, \quad b_t=\sum_{i=1}^{t}\beta^{\,t-i} g_i.$
>
> Hence, for every step $t$, $b_t=\frac{1}{1-\beta}m_t.$
>
> So when $\beta=0.95$, the Spectra buffer is indeed $20\times$ larger in magnitude, but every term in the momentum sum is scaled by the same factor $20$. Therefore, this does not mean that Spectra places relatively more emphasis on the current gradient than Muon; the two buffers are strictly collinear at every step.
>
> In our experiments, this constant-factor difference does not affect the actual parameter update, because we explicitly match the final update RMS norm at every step. Under this protocol, a global rescaling of the momentum buffer is absorbed by the subsequent RMS normalization/rescaling, so the optimizer sees the same update direction and the same final update magnitude. In this sense, the two implementations are equivalent in our setting.
>
> *Q2: I am curious whether the experiments in Section 4.1 used matched learning rates across AdamW, Muon, Dion, and Spectra, or whether update RMS was equalized as a fairer comparison baseline, to avoid confounds from hyperparameter differences.*
>
> **A2:** In Section 4.1, we equalized the effective update RMS across AdamW, Muon, Dion, and Spectra. Concretely, we kept the learning-rate schedule fixed and matched the per-step update magnitude across the four settings, so the comparison isolates the effect of update direction, rather than confounding it with different step scales.
>
> Our design follows the standard RMS-alignment practice used in prior Muon work. Specifically, prior large-scale Muon studies [1] report that AdamW’s update RMS is typically around 0.2~0.4, and therefore rescale Muon updates into that range so that AdamW-tuned learning rates and weight decay can be reused directly. The Kimi K2 technical report [2] adopts the same principle: it explicitly describes “consistent update RMS scaling”, and its MuonClip algorithm rescales the orthogonalized update by $\sqrt{max(n,m)} * 0.2$ to “Match Adam RMS.”
>
> Following this convention, in our implementations of Spectra and Dion we use the same RMS-matching protocol as well: all optimizers share the same learning-rate schedule and the same effective update RMS, while differing only in how the update direction is transformed—element-wise for AdamW, orthogonalized for Muon/Dion, and spectrally modulated for Spectra. Therefore, the gains reported in Section 4.1 should be attributed to the optimizer’s update geometry rather than to mismatched hyperparameter scales.
>
> [1] https://arxiv.org/abs/2502.16982
>
> [2] https://arxiv.org/abs/2507.20534

---

> > ### Author Rebuttal · Reviewer_Fwmz · 2026-04-02
> >
> > Thank you to the authors for their additional clarifications which were helpful. Given the high quality of both the work and the response, I will maintain my initial score.

---

> > > ### Author Response · Authors · 2026-04-07
> > >
> > > Thank you for taking the time to review our rebuttal. We appreciate your consideration.

---

### Official Review · Reviewer_jJ3M · 2026-03-09

**Soundness:** 3
**Presentation:** 3
**Significance:** 2
**Originality:** 3
**Overall Recommendation:** 4
**Confidence:** 3

**Summary:**

The paper argues that LLM gradients are highly anisotropic in spectral space: a very small set of top singular directions (“spike,” about 1.5% of directions) carries most of the gradient energy, while the remaining directions form a long “tail.” The authors claim this spike mainly corresponds to common linguistic structure, while the tail carries rarer, context-specific semantic information.

To address this, the paper proposes Spectra, a spike-aware optimizer. Instead of flattening the whole spectrum, Spectra only suppresses the dominant low-rank spike subspace and leaves the tail mostly unchanged.
Experimentally, on Qwen3-0.6B (100B tokens) and LLaMA3-8B (50B tokens), the paper reports that Spectra achieves faster convergence, lower final loss, better downstream accuracy.

**Compliance With Llm Reviewing Policy:**

Affirmed.

**Final Justification:**

Some of my minor concerns are unsolved. Overall, I maintain my score.

**Key Questions For Authors:**

- In your Figure 5, I think this problem arises from the bf16 matmul in NS itertaion? When you replace it with fp32, this may not happen since NS iteration is mathematically persevering the singular spaces. Is that true? I also want to see how much improvement there is if you use fp32 NS iteration in Muon. I think this can provide stronger evidences for your section 2.5.
- In your Table 5, why does increasing T lead to degradation performance? I think it is a similar numerical problem, just like Figure 5.
- How universal is the 1.5% spike ratio? Does this remain true for much larger models and different data? Or what is the scaling trend?
- Why is shrink-to-tail-average the right operation? Why do you not shrink it to \sigma_max(M_tail) or anything else?
- Is the spike always harmful? If the spike captures common structure, suppressing it too much might hurt learning of useful shared patterns. How do you think of it?

**Limitations:**

Unfortunately, the authors did not talk about limitations. See weakness and questions above.

**Strengths And Weaknesses:**

Strength:
- the idea is clear: do not flatten everything; only suppress the dominant spike. The story sound kind of reasonable.
- I like some of analyses in the paper, including Figure 4 and 3.
- ablation study on introduced hyperparameters are good
- the experiment scale, i.e., 8B model is good.

Weakness:
- The semantic interpretation may be too strong. The paper claims the spike mostly reflects common grammar/frequency structure and the tail reflects fine-grained semantics. The interventions support this direction, but the evidence still feels indirect, not fully conclusive
- Some design choices feel heuristic. The default 1.5% spike rank, the shrink-to-tail-average rule, and the RMS rescaling are reasonable, but still somewhat heuristic. The paper could better explain why these are the right choices.
- There is no empirical evidences about how tight the theory is.

---

> ### Author Rebuttal · Authors · 2026-03-31
>
> Thank you very much for your thoughtful comments. Our responses are as follows:
>
> *Q1: The semantic evidence feels not fully conclusive.*
>
> **A1:** In a highly entangled nonlinear system such as an LLM, it is difficult to establish a fully clean causal decomposition showing that spike directions are strictly grammar/frequency while tail directions are strictly semantics.
>
> What our experiments support is a strong statistical association: FreqNorm and Shuffle mainly perturb the spike singular values, while leaving the tail much less affected. In such a complex system, this highly localized spectral response is strong evidence that the spike is more associated with shared/common structures such as frequency and syntax, whereas the tail is more sensitive to fine-grained, context-dependent semantics.
>
> We will revise related statements, if overly strong, to present the findings accurately, replacing absolute terms such as “is driven by” with more precise statistical language such as “is more associated with.”
>
> *Q2: Some design choices feel heuristic, e.g. the default 1.5% spike rank and the RMS rescaling.*
>
> **A2:** For the rank ratio $r$, the choice is not highly scale-specific. In Figure 1 of the main paper and Figures 9–11 in Appendix A.1, we analyze gradient anisotropy across different layers and modules from 0.6B up to 32B models, and consistently observe that the spike subspace occupies only about 1.5% of directions.
>
> For the RMS rescaling, our design follows the standard RMS-alignment practice used in prior Muon work. Specifically, prior large-scale Muon studies [1] report that AdamW’s update RMS is typically around 0.2–0.4, and therefore rescale Muon updates to 0.2. The Kimi K2 technical report [2] adopts the same principle.
>
> *Q3: There is no empirical evidence about how tight the theory is.*
>
> **A3:** Equation (1) is derived from the standard second-order Taylor expansion of the loss. The key bound in Equation (2) relies on the spike-dominated sample noise. Empirically, the parameter updates exhibit strong anisotropy, as shown in Figure R1 [3]. As a result, the covariance of sample-wise gradients is also highly anisotropic, with the spike region dominating most of the covariance energy. This is exactly the regime in which the bound becomes relatively tight.
>
> *Q4: In Figure 5, this problem arises from the bf16 matmul in NS? What about fp32?*
>
> **A4:** The official Muon implementation uses bf16 for the NS iteration. We also provide an fp32 control experiment; with fp32 NS iteration, all singular directions are indeed better preserved, as shown in Figure R2 [3]. However, if we switch to fp32 NS iteration throughout training, the computational cost becomes nearly 2×. The measured latency is shown below:
>
> **Empirical latency comparison (ms) on H200 GPU**
>
> |Matrix Size|NS-bf16|NS-fp32|Power-Iter T=1|
> |-|-:|-:|-:|
> |(4096,4096)|3.6193|7.1236|0.9673|
> |(4096,14336)|9.3501|19.8029|1.7766|
>
> *Q5: In Table 5, why does increasing T lead to degraded performance?*
>
> **A5:** Increasing  $T$ does not progressively perturb the tail directions. The effect of power iteration is mainly confined to the spike subspace, where it refines the estimated singular directions, while the tail singular components are largely preserved, as shown in Figure R3 [3]. The exact reason why a larger $T$ still leads to degraded downstream performance remains unclear, and we leave it to future work.
>
> *Q6-7: Why is shrink-to-tail-average the right operation? Is the spike always harmful?*
>
> **A6-7:** The spike is not always harmful. On the contrary, it contains important common structure. The issue is not the existence of the spike itself, but its excessively large scale. The goal is not to remove the spike, but to rescale it to a more appropriate level.
>
> We ablated several spike treatments: spike-min (replacing spike with the min singular value of spike), zero-spike (hard removal), soft-clipping (continuous shrinkage schemes), and ours (tail-average replacement). Two conclusions are clear: (1) suppressing spike scale is fairly robust—multiple variants improve over leaving it unchanged; (2) zeroing the spike is clearly worse than rescaling it, showing that spike directions still contain useful signal and should not be discarded entirely.
>
> **Downstream performance under different spike-value treatments**
>
> |Model|Spike Treatment|Loss|ArcC|ArcE|HellaSwag|LAMBADA|PIQA|RACE|Avg|
> |-|-|-:|-:|-:|-:|-:|-:|-:|-:|
> |Qwen3-0.6B|Spike-min|3.06|26.62|49.07|43.27|40.66|68.93|49.41|46.33|
> |Qwen3-0.6B|Zero-spike|3.07|24.32|46.17|37.98|34.43|66.92|49.15|43.16|
> |Qwen3-0.6B|Soft-clipping|**3.00**|25.26|49.16|**43.38**|**41.65**|**69.64**|**49.47**|46.43|
> |Qwen3-0.6B|Ours|3.01|**26.79**|**49.92**|42.84|41.53|69.04|49.19|**46.55**|
>
> *Q8: Limitation.*
>
> **A8:** Sorry for the character-limit constraint. Please see Reviewer 9tL3, A8.
>
> [1] https://arxiv.org/abs/2502.16982
>
> [2] https://arxiv.org/abs/2507.20534
>
> [3] https://anonymous.4open.science/r/plots-F802

---

> > ### Author Rebuttal · Reviewer_jJ3M · 2026-04-02
> >
> > Thank you for the rebuttal. I feel that Q1, Q3, and Q5 are still not fully resolved.
> >
> > For Q3, the second inequality in Eq. (2) and the first inequality in Eq. (3) are not sufficiently justified.
> >
> > For Q5, I think this is a very interesting question and would make a valuable addition to the paper. I am still confused about what Figure R3 is meant to show. In power iteration, do we only recover the top-$k$ singular subspaces? If so, why does the curve extend to the last singular index? Could you provide more clarification or a more detailed caption for this figure?
> >
> > For Q1, I am fine with the authors revising the writing. That said, providing further evidence would still be interesting and valuable.
> >
> > Overall, I will keep my score.

---

> > > ### Author Response · Authors · 2026-04-07
> > >
> > > Thank you very much for your careful review. We summarize our responses as follows:
> > >
> > > **Q1:** *For Q1, I am fine with the authors revising the writing. That said, providing further evidence would still be interesting and valuable.*
> > >
> > > **A1:** Thank you, we will treat the more direct semantic analysis of the spike/tail decomposition in model representations and gradients as valuable future work.
> > >
> > > **Q3:** *For Q3, the second inequality in Eq. (2) and the first inequality in Eq. (3) are not sufficiently justified.*
> > >
> > > **A3:** In a complex system such as an LLM, it is inherently difficult to impose sufficiently strong and fully verifiable assumptions on the Hessian. For the first inequality in Eq. (3), we use the assumption $\mathbf{H}\succeq \mu \mathbf{I}$ for some $\mu>0$, which is a standard lower-curvature / strong-convexity-type assumption in Hessian-based optimization analysis. In the twice-differentiable case, this type of assumption is commonly used to express a lower bound on curvature [1].
> > >
> > > For the second inequality in Eq. (2), in the current simplified form, we drop the preceding Hessian term $\bar{\mathbf{g}}^\top \mathbf{H}\bar{\mathbf{g}}$ in the denominator. A tighter version can in fact be obtained under the same assumption $\mathbf{H}\succeq \mu \mathbf{I}$ by keeping this term explicitly, rather than removing it, which makes the bound more justified and tighter.
> > >
> > > Empirically, the usefulness of Theorem 2.1 is supported by the learning-rate ablation in our Figure 8: Spectra can stably use a larger learning rate than Adam and achieve better convergence loss, which is consistent with the theorem’s qualitative prediction.
> > >
> > >
> > > **Q5:** *For Q5, I think this is a very interesting question and would make a valuable addition to the paper. I am still confused about what Figure R3 is meant to show. In power iteration, do we only recover the top-$k$ singular subspaces? If so, why does the curve extend to the last singular index? Could you provide more clarification or a more detailed caption for this figure?*
> > >
> > > **A5:** Figure R3 uses the same data as Figure 5 in the paper, but applies the full Spectra shaping operation. Concretely, we first use cached Power-Iteration SVD with different iteration counts to estimate the spike subspace, and then rescale the recovered spike singular values to match the tail RMS, following the Spectra optimizer.
> > >
> > > Figure R3 visualizes how well the singular-vector directions of the shaped update are preserved relative to the original update. Power-Iteration indeed only recovers the top-$k$ singular subspaces. The main point of Figure R3 is that the resulting approximation error is largely confined to the spike region and does not significantly propagate to the tail singular directions.
> > >
> > > [1] https://arxiv.org/abs/1606.04838

---

### Official Review · Reviewer_9tL3 · 2026-03-10

**Soundness:** 2
**Presentation:** 2
**Significance:** 2
**Originality:** 3
**Overall Recommendation:** 4
**Confidence:** 4

**Summary:**

The paper investigates the anisotropic spectral properties of gradient signals during Large Language Model (LLM) pretraining. The authors observe a consistent "low-rank spike + smooth tail" structure in the gradient singular spectrum across multiple model scales and training stages. They theoretically and empirically demonstrate that traditional optimizers like AdamW are dominated by this low-rank spike. This dominance suppresses the learning of long-tail semantic features by contracting tail updates through second-moment normalization and tightly bounding the globally optimal learning rate.

To address this, the authors introduce Spectra, a spike-aware optimizer. Spectra uses a cached, warm-started power iteration method to efficiently track the low-rank spike subspace. The authors demonstrate that aggressively flattening the entire spectrum can amplify numerical noise in the fragile spectral tail. Therefore, instead of global flattening, Spectra selectively shrinks the spike's singular values toward the tail's average scale. Evaluated on Qwen3-0.6B and LLaMA3-8B architectures, Spectra demonstrates faster wall-clock convergence, a reduced optimizer-state memory footprint, and improved downstream benchmark accuracy compared to AdamW and Muon.

**Compliance With Llm Reviewing Policy:**

Affirmed.

**Final Justification:**

Overall, the rebuttal addressed my previous concerns well, and the new larger-scale results make the paper’s main claims more convincing. My overall view of the paper is now more positive.

At the same time, I hope the authors will include these new analyses and experimental results in the revised version, so that the paper itself matches what was clarified in the rebuttal. In particular, the mix-up in the original draft between results from different model scales should be corrected clearly and carefully, so that the abstract, introduction, and experiments do not give readers the wrong impression.

**Key Questions For Authors:**

See the detailed weaknesses above.

**Limitations:**

The authors should include a dedicated Limitations section or append one to the conclusion.

**Strengths And Weaknesses:**

>**Disclosure (Policy B):** In compliance with the ICML 2026 LLM Policy B (Permissive), to improve linguistic accuracy and communication efficiency, I used a LLM to translate and polish this review. All opinions, technical analyses, and judgments were written by me in my native language; the LLM was used only for translation and language refinement.

**Strengths**

The work focuses on a practically important problem: addressing the highly anisotropic nature of gradient signals to enable memory-efficient and faster pretraining of large language models. By creatively combining cached, warm-started power iteration with localized spike shrinkage, Spectra effectively targets the dominant low-rank subspace without amplifying noise in the spectral tail. The empirical validation across Qwen3 models (up to 32B parameters) successfully confirms a consistent "low-rank spike + smooth tail" profile, and the linguistic ablation studies cleanly link these gradient dimensions to syntactic and semantic dataset properties. Spectra achieves a 49.25% reduction in optimizer-state memory and reaches the target loss 30% faster than AdamW during the pretraining of a LLaMA3-8B model on 50 billion tokens, which is a highly compelling practical result.

**Weaknesses**

- Theorem 2.1 analyzes a vanilla SGD update, while both AdamW and Spectra use substantially different update rules involving momentum and normalization; Spectra further applies spectral shaping and RMS rescaling. Therefore, the theorem mainly provides qualitative motivation rather than a direct analysis of the proposed optimizer.
- The authors claim in the abstract, introduction, and experimental results that Spectra reaches the same target loss 30% faster than AdamW in wall-clock time as a primary contribution. However, the training loss curves in Figure 7 fail to demonstrate this acceleration. Furthermore, there is a textual inconsistency regarding this claim. The abstract states that on LLaMA3-8B trained on 50B tokens, Spectra reaches the same target loss 30% faster than AdamW. Conversely, the Main Results section asserts that on the 0.6B model, Spectra achieves a final validation loss that is 2.1% lower than AdamW and 1.4% lower than Muon, and reaches a matched loss level 30% faster in wall-clock time than AdamW. This contradictory presentation causes reader confusion.
- The evaluation of computational efficiency is restricted to the 0.6B model scale. This limited scope fails to substantiate the claim made in the introduction that Spectra is well suited for large-scale parallel and distributed training without requiring full-gradient synchronization. In practice, large-scale distributed training typically involves a complex hybrid of data, tensor, pipeline, and context parallelism. Therefore, extending the efficiency analysis to larger configurations would provide more convincing support for the claims regarding large-scale distributed training compatibility.
- While the motivation is novel, the actual intervention of replacing the spike singular values with the average tail scale $\sigma_{tail}$ is only one specific design choice. The paper does not compare this choice against alternatives such as soft clipping, adaptive damping, or different shrinkage targets. As a result, it is unclear whether this particular form is especially well justified, or simply one of several viable options in a broader design space.
- The impact may be slightly limited by the introduction of new hyperparameters (rank ratio $r$ and power iterations T), though the ablation studies suggest the method is relatively robust to these choices at the current tested scales. However, it remains to be seen whether these hyperparameters require careful tuning at larger scales or different architectures.
- All experiments use only the dense architecture and DCLM dataset for pretraining. It is unclear whether the spike and tail structure and Spectra's benefits persist on different data distributions or architectures (e.g., code, multilingual corpora, instruction-tuning data, or MoE architectures).
- Only two model scales are tested, and the larger model (8B) is trained on only 50B tokens, which is far below typical practice.

---

> ### Author Rebuttal · Authors · 2026-03-31
>
> Thank you very much for your thoughtful comments. Our responses are as follows:
>
> *Q1: Theorem 2.1 is stated for SGD and seems weakly connected to Adam/Muon/Spectra.*
>
> **A1:** Theorem 2.1 is meant to capture a general principle, not an SGD-specific claim: **spike-dominated sample noise limits the optimal learning rate**.
>
> SGD form is only for simplicity. More generally, the same argument applies to any update $\mathbf{w}^+=\mathbf{w}-\eta\mathbf{g}$, where $\mathbf{g}$ is the *effective update direction* (raw gradient for SGD, optimizer-transformed update for Adam/Muon/Spectra). Under this view, the same bound in equation (3) still holds.
>
> So as long as the actual optimizer update remains spike-dominated, the same learning-rate constraint applies. Empirically, Adam’s element-wise scaling does not remove this anisotropy, as shown in Figure R1 [1], which motivates spectral-space corrections such as Muon, Spectra.
>
> *Q2: The 30% wall-clock speedup is not clearly supported by Fig. 7, and the abstract/main text appear to mix 8B and 0.6B results.*
>
> **A2:** In the original submission, the detailed experimental analyses were conducted on the 0.6B model due to time and resource constraints. The abstract’s reference to larger-scale wall-clock speedup was an unintended oversight, the abstract will be updated accordingly to accurately reflect the evaluated scale.
>
> Regarding time-to-target wall-clock results, Section 4.1 shows that Spectra and Muon achieve approximately 30% and 25% speedup (see below) over AdamW on average, respectively. As reported in prior work [2], though Muon is more data-efficient than AdamW, its spectral-level matrix operations increase computational overhead at larger scales, diminishing wall-clock advantage. At the 8B scale (see below), Muon’s wall-clock performance becomes comparable to or even worse than AdamW. Spectra is designed to address this scaling limitation. While its speedup over AdamW decreases to approximately 16% at the 8B scale, it maintains a clear advantage over Muon by reducing the cost of spectral operations.
>
> **Qwen3-0.6B time-to-target**
>
> |Loss|AdamW steps|AdamW h|Spectra steps|Spectra h|Spectra spd|Muon steps|Muon h|Muon spd|
> |-|-:|-:|-:|-:|-:|-:|-:|-:|
> |3.7|4955|7.29|**3575**|**5.22**|**1.40×**|3950|5.91|1.23×|
> |3.5|7290|10.73|**5240**|**7.65**|**1.40×**|5250|7.85|1.37×|
> |3.3|11084|16.31|**9050**|**13.22**|**1.23×**|9500|14.21|1.15×|
> |3.0|50000|73.57|**38339**|**56.01**|**1.31×**|38951|58.26|1.26×|
>
> **LLaMA3-8B time-to-target**
>
> |Loss|AdamW steps|AdamW h|Spectra steps|Spectra h|Spectra spd|Muon steps|Muon h|Muon spd|
> |-|-:|-:|-:|-:|-:|-:|-:|-:|
> |3.7|1632|7.89|**1440**|**6.88**|**1.15×**|1660|8.27|0.95×|
> |3.5|2120|10.25|**1861**|**8.89**|**1.15×**|2140|10.66|0.96×|
> |3.3|2850|13.77|**2520**|**12.03**|**1.14×**|2745|13.67|1.01×|
> |3.0|5750|27.79|**5000**|**23.88**|**1.16×**|5133|25.56|1.09×|
> |2.8|10090|48.75|**9100**|**43.46**|**1.12×**|9100|45.31|1.08×|
>
> *Q3: The efficiency evaluation at larger-scale is needed.*
>
> **A3:** Under tensor-parallel partitioning, PowerIter only requires local matrix--thin-matrix multiplications plus an All-Reduce on a $r$-rank thin matrix and a small $r$-dimensional vector.
>
> We provide explicit 8B evidence. Table 2 in paper reports operator-level latency on representative 8B gradient shapes, where Power-Iter is consistently much faster than NS. At the end-to-end level, the measured per-step time on the 8B model is 17191 ms for Spectra, compared to 17398 ms for AdamW and 17929 ms for Muon; the 8B time-to-target results further show about **1.15×–1.16×** speedup over AdamW.
>
> *Q4: Spike treatments ablation.*
>
> **A4:** Sorry for the character-limit constraint. It show that suppressing the spike scale is fairly robust. Please see Reviewer jJ3M, A6-7.
>
> *Q5: Rank ratio and power iterations require careful tuning at larger scales?*
>
> **A5:** For the rank ratio, the choice is not highly scale-specific. In Figure 1 of main paper and Figures 9–11 in Appendix A.1, we analyze gradient anisotropy across layers and modules from 0.6B up to 32B models, and consistently observe that the spike subspace occupies only about 1.5% directions.
>
> For the number of power iterations $T$, 1 is enough. This relies on the strong consistency of the spike directions across neighboring steps, so 1 warm-started power iteration provides a good approximation.
>
> Ablation results at larger scales will be incorporated into the experimental section.
>
> *Q6-7: Spectra's benefits on different data distributions, model scales, training stages or architectures.*
>
> **A6-7:** Sorry for the character-limit constraint. Spectra's generalizability is confirmed on MoE architecture and code data. Please see Reviewer BHSx, A3.
>
> *Q8: Limitation.*
>
> **A8**: Due to time and resource constraints, Spectra has not yet been comprehensively validated or ablated across diverse data distributions, model scales, and architectures.
>
> [1] https://anonymous.4open.science/r/plots-F802
>
> [2] https://arxiv.org/abs/2502.16982

---

> > ### Author Rebuttal · Reviewer_9tL3 · 2026-04-02
> >
> > I thank the authors for their detailed rebuttal. The time-to-target tables for both 0.6B and 8B are helpful additions. However, I maintain my original score for the following reasons:
> >
> > Scalability concern: The newly provided data reveals that the wall-clock speedup diminishes from \~30% at 0.6B to \~15% at 8B, and the per-step time difference at 8B is marginal (~1.2%). This downward trend raises questions about whether the benefits will persist at truly large scales, which weakens the paper’s central claim of being “well suited for large-scale parallel and distributed training.” The claimed advantages for distributed training settings also remain unsubstantiated.
> >
> > Theoretical gap: The response to Q1 does not provide a formal extension of Theorem 2.1 to Adam-family optimizers. The claim that “the same bound still holds” for arbitrary optimizer updates requires more rigorous justification.
> >
> > I acknowledge the authors’ honest corrections and the practical value of the optimizer-state memory reduction. However, the above concerns remain substantive, and I maintain my score of 3.

---

> > > ### Author Response · Authors · 2026-04-07
> > >
> > > Thank you very much for your careful review. We summarize our responses as follows:
> > >
> > > *Q1: Scalability concern.*
> > >
> > > **A1:** Regarding wall-clock speedup, we additionally evaluate Spectra on Qwen3-2B-A0.8B and Qwen3-8B, where it achieves over 24% speedup in both cases, indicating that the smaller gain on LLaMA3-8B is architecture-dependent. In end-to-end per-step overhead measurements on Qwen3-0.6B to 32B, the runtime ordering is consistently *Spectra < Adam < Muon*. Spectra is therefore lighter than Adam while converging faster than Muon in wall-clock time, reducing overall training cost by more than 20% in large-scale pretraining.
> > >
> > > The wall-clock results on Qwen3 2B-A0.8B and Qwen3-8B are showed below: Spectra achieves a 34% speedup over Adam on the former and 24% on the latter. Spectra’s speedup advantage over Muon also grows with scale, increasing from 4% at 0.6B to 6% at Qwen3 2B-A0.8B and 8.7% at Qwen3-8B. Importantly, Muon’s stronger data efficiency than Adam has already been validated at much larger scales, e.g., in Moonlight 3B/16B MoE training on 5.7T tokens [1] and Kimi K2 1T MoE on 15.5T tokens [2]. Therefore, since Spectra consistently outperforms Muon in our experiments, it is expected to preserve a consistent wall-clock advantage over Adam as model scale increases.
> > >
> > > **Qwen3-2B-A0.8B time-to-target**
> > > |Loss|AdamW steps|AdamW h|Spectra steps|Spectra h|Spectra spd|Muon steps|Muon h|Muon spd|
> > > |---|---:|---:|---:|---:|---:|---:|---:|---:|
> > > |3.7|2030|2.75|**1600**|**2.18**|**1.26×**|1750|2.41|1.14×|
> > > |3.5|3106|4.21|**2486**|**3.39**|**1.24×**|2579|3.55|1.19×|
> > > |3.3|6000|8.13|**4436**|**6.05**|**1.34×**|4678|6.45|1.26×|
> > >
> > > **Qwen3-8B time-to-target**
> > > |Loss|AdamW steps|AdamW h|Spectra steps|Spectra h|Spectra spd|Muon steps|Muon h|Muon spd|
> > > |---|---:|---:|---:|---:|---:|---:|---:|---:|
> > > |3.7|3400|9.58|**2957**|**8.01**|**1.20×**|3696|10.52|0.91×|
> > > |3.5|4400|12.40|**3829**|**10.37**|**1.20×**|4133|11.76|1.05×|
> > > |3.3|6327|17.83|**5164**|**13.99**|**1.27×**|5651|16.08|1.11×|
> > > |3.1|9903|27.90|**8321**|**22.54**|**1.24×**|8612|24.50|1.14×|
> > >
> > > Spectra is also more lightweight than Adam. The table below reports end-to-end per-step runtime from Qwen3-0.6B to 32B. Across these scales, Spectra consistently achieves the lowest per-step overhead among the three optimizers.
> > >
> > > **Qwen3 per-step runtime**
> > >
> > > |Model|AdamW (ms)|Muon (ms)|Spectra (ms)|
> > > |---|---:|---:|---:|
> > > |Qwen3-0.6B|5297.0|5385.0|**5259.0**|
> > > |Qwen3-2B-A0.8B|4312.9|6064.7|**3973.7**|
> > > |Qwen3-8B|10143.0|10243.0|**9750.0**|
> > > |Qwen3-32B|16308.0|16690.5|**16275.9**|
> > >
> > >
> > > *Q2: Theoretical gap.*
> > >
> > > **A2:** **General theorem.** We provide an optimizer-agnostic extension of Theorem 2.1. Consider a one-step update $\mathbf{w}^{+}=\mathbf{w}-\eta\boldsymbol{\phi}$, where $\boldsymbol{\phi}$ is the optimizer update direction, and define $\mathbf{m}\_{\phi}=\mathbb{E}[\boldsymbol{\phi}]$ and $\mathbf{Q}\_{\phi}=\mathbb{E}[\boldsymbol{\phi}\boldsymbol{\phi}^{\top}]$. A second-order expansion gives
> > > $L(\mathbf{w}-\eta\boldsymbol{\phi}) \approx L(\mathbf{w})-\eta\mathbf{g}^{\top}\boldsymbol{\phi}+\frac{1}{2}\eta^{2}\boldsymbol{\phi}^{\top}\mathbf{H}\boldsymbol{\phi},$ so the mean-optimal learning rate is $\eta_{\phi}^{\star}=\frac{\mathbf{g}^{\top}\mathbf{m}_{\phi}}{\operatorname{tr}(\mathbf{H} \mathbf{Q}\_{\phi})}$.
> > >
> > > This recovers Theorem 2.1 as the SGD special case $\boldsymbol{\phi}=\mathbf{g}\_{B}$, where $\mathbf{Q}\_{\phi}=\mathbb{E}[\mathbf{g}\_{B}\mathbf{g}\_{B}^{\top}]=\mathbf{g}\mathbf{g}^{\top}+\frac{1}{B}\mathbf{\Sigma}$.
> > >
> > > For AdamW, $\boldsymbol{\phi}$ is the actual preconditioned update:
> > > $\boldsymbol{\phi}\_{t}=\frac{\widehat{\mathbf{m}}\_{t}}{\sqrt{\widehat{\mathbf{v}}\_{t}}+\epsilon}.$ The same result applies with $\mathbf{m}\_{\phi}=\mathbb{E}[\boldsymbol{\phi}\_{t}]$ and $\mathbf{Q}\_{\phi}=\mathbb{E}[\boldsymbol{\phi}\_{t}\boldsymbol{\phi}\_{t}^{\top}]$, yielding $\eta\_{\phi}^{\star}=\frac{\mathbf{g}^{\top}\mathbb{E}[\boldsymbol{\phi}\_{t}]}{\operatorname{tr}(\mathbf{H}\,\mathbb{E}[\boldsymbol{\phi}\_{t}\boldsymbol{\phi}\_{t}^{\top}]}.$
> > >
> > > **Why Adam's element-wise normalization does not remove the spike.** Adam rescales coordinates independently, but does not remove dense cross-coordinate correlations. Since the spike is a directional low-rank structure rather than a few isolated large coordinates, Adam may reduce its magnitude but does not eliminate the spike itself.
> > >
> > > **Empirical evidence.** For AdamW, the across-example covariance spectrum of per-example updates exhibits a clear spike-plus-tail structure, with a few leading eigenvalues capturing most of the variance, as shown in Figure R4 [3]. Therefore, under Adam, the optimal learning rate is constrained by spike-dominant update variance. This matches the learning-rate sweep in Figure 8, where Spectra supports a larger stable and effective learning-rate region than Adam.
> > >
> > > [1] https://arxiv.org/abs/2502.16982
> > >
> > > [2] https://arxiv.org/abs/2507.20534
> > >
> > > [3] https://anonymous.4open.science/r/plots-F802/discuss_figures.pdf

---

### Official Review · Reviewer_BHSx · 2026-03-10

**Soundness:** 4
**Presentation:** 3
**Significance:** 4
**Originality:** 3
**Overall Recommendation:** 5
**Confidence:** 5

**Summary:**

This paper introduces Spectra, a novel optimizer designed for Large Language Model (LLM) training. The authors first conduct an analysis of gradient anisotropy using Singular Value Decomposition (SVD), empirically demonstrating that the gradient spectrum splits into two distinct parts: a massive "spike" driven by common linguistic structures, and a "smooth tail" that encodes fine-grained, context-dependent semantics and rare world knowledge. The paper reveals that this spike dominance suppresses long-tail learning through two coupled mechanisms: (1) locally, spike-dominated second-moment accumulation artificially shrinks tail updates; and (2) globally, spike-dominated stochastic variance severely restricts the maximum stable learning rate. Furthermore, the authors demonstrate that tail directions are highly noise-dominated, meaning that full-spectrum iterative approximation methods (like Newton-Schulz) actively damage long-tail learning by scrambling these fragile semantic signals. Based on these findings, the authors propose Spectra. Instead of flattening the entire gradient spectrum, Spectra selectively shrinks only the problematic spike while leaving the delicate tail completely untouched. This targeted intervention successfully balances the update steps and removes the need for dense second-moment tracking, significantly saving memory and accelerating training. Experimental results confirm that Spectra delivers faster convergence and superior downstream performance compared to standard baselines.

**Compliance With Llm Reviewing Policy:**

Affirmed.

**Final Justification:**

The response has solved my questions, I keep the current score.

**Key Questions For Authors:**

Question 1: Regarding the FreqNorm and Shuffle interventions in Figure 2, to what extent are the changes in the singular spectrum driven by structural shifts versus global changes in loss magnitude? Specifically, since FreqNorm rescales token-wise losses, it likely reduces the overall loss magnitude, which could naturally lead to smaller singular values. Conversely, shuffling sentences likely increases the overall loss, thereby inflating the gradient magnitude. Have you verified whether simply scaling the standard loss by a constant (without token-wise weighting) produces a similar trend to FreqNorm?

Question 2: Regarding computational efficiency, the text highlights that Spectra is "5.1× faster in optimizer processing time" compared to Muon. However, looking at table 3, the difference between the optimizers is much smaller. Could you explicitly clarify the relationship between the isolated "optimizer processing time" and the overall step time?

**Limitations:**

yes

**Strengths And Weaknesses:**

Strengths:

1. Novel Insights: The paper introduces the compelling insight that the long-tail singular directions of the gradient encode fine-grained, context-dependent semantics and rare world knowledge. Furthermore, it crucially identifies that this tail is highly noise-dominated, providing a strong theoretical explanation for why existing optimizers struggle with rare facts.
2. Rigorous Empirical Validation: The authors do not rely solely on theory; their core observations and assumptions regarding gradient anisotropy are thoroughly and creatively validated through targeted ablation experiments (e.g., the token-shuffling and frequency-normalization tests).
3. Highly Efficient Algorithmic Design: The proposed Cached Power-Iteration SVD is a brilliant engineering solution. By exploiting the temporal continuity of the gradient's structural "spike," the algorithm successfully reduces the required power iterations to just one step, drastically cutting computational overhead.
4. Superior Empirical Performance: The approach delivers faster convergence and better average downstream performance across standard benchmarks compared to both AdamW and the recent spectral optimizer, Muon.

Weakness:

1. A primary limitation of the empirical evaluation is the restricted pretraining tokens. Training an 8-billion parameter model on only 50 billion tokens represents the early stages of a standard modern LLM pre-training run. It remains unclear how the optimizer behaves as training approaches full saturation even the convergence is faster at the earlier stage.

---

> ### Author Rebuttal · Authors · 2026-03-30
>
> Thank you very much for your time and valuable comments. Our responses are as follows:
>
> *Q1: It remains unclear whether the observed changes in singular values stem from structural variations or from changes in the loss function.*
>
> **A1:** The loss scale can affect the overall magnitude of the gradient. If the effect were purely due to loss scaling, then the gradient matrix would be approximately rescaled as $G' \approx cG$, and the singular values would change almost uniformly as $\sigma_i(G') \approx |c|\,\sigma_i(G)$. In that case, the main effect would be a global scaling of the spectrum, rather than a structural redistribution across different singular directions.
>
> What we observe in the paper is not a uniform rescaling but a *highly localized spectral response*. In particular, the FreqNorm and Shuffle interventions mainly perturb the spike region, while leaving much of the tail comparatively less affected. In such a complex model, this kind of localized response is strong evidence that the spike subspace is closely related to shared/common structures such as frequency and syntax.
>
> *Q2: Clarify "optimizer processing time" and "the overall step time".*
>
> **A2:** *Optimizer processing time* refers only to the optimizer-internal parameter-update computation after gradients are available, whereas *overall step time* measures the full end-to-end training iteration (forward + backward + optimizer update).
>
> Therefore, the statement “5.1× faster in optimizer processing time” refers specifically to the optimizer-side computation reported in Table 2, not to the end-to-end step time in Table 3. In Muon, this cost is dominated by the Newton–Schulz iteration; in Spectra, it is dominated by the power iteration used to track the spike subspace. Under the practical gradient shapes arising in our 8B setup, Spectra reduces this optimizer-side cost by about 5.1× relative to Muon.
>
> By contrast, the overall step time in LLM training is dominated by the forward and backward passes, especially the large GEMMs and attention kernels over the token dimension. As a result, even a substantial reduction in optimizer-side cost leads to only a modest change in end-to-end step time, which explains why Table 3 shows much smaller differences.
>
> *Q3: Insufficient tokens for training the 8B model.*
>
> **A3:** Due to limited computational resources, the additional experiments we provide in the rebuttal were conducted on both Qwen3-0.6B and a MoE architecture, Qwen3-2B-A0.8B (based on Qwen3-235B-A22B [1] and scaled down to 2B-A0.8B).
>
> To further validate the applicability of Spectra, we first pretrain a Qwen3-2B-A0.8B MoE model on DCLM up to 20B tokens. The downstream results are shown below. Spectra achieves the best average downstream performance.
>
> | Model | Optimizer | Loss | ArcC | ArcE | HellaSwag | LAMBADA | PIQA | RACE | Avg |
> |---|---|---:|---:|---:|---:|---:|---:|---:|---:|
> | Qwen3-2B-A0.8B | AdamW   | 3.03 | 24.57 | 43.48 | 37.68 | 27.83 | 65.13 | 47.81 | 41.08 |
> | Qwen3-2B-A0.8B | Muon    | **2.96** | 25.51 | 47.14 | **43.15** | 33.44 | **68.99** | 47.12 | 44.23 |
> | Qwen3-2B-A0.8B | Spectra | 2.97 | **27.05** | **48.70** | 43.04 | **34.66** | 68.06 | **48.68** | **45.03** |
>
> We also evaluate Spectra in the code domain by training Qwen3-0.6B on the RedPajama dataset [2] up to 20B tokens and testing on the code-generation benchmark MBPP, where the result is **Spectra (17.4) > AdamW (16.6) > Muon (14.4)**. Spectra again achieves the best overall performance.
>
> [1] https://huggingface.co/Qwen/Qwen3-235B-A22B
>
> [2] https://huggingface.co/datasets/togethercomputer/RedPajama-Data-1T

---

> > ### Author Rebuttal · Reviewer_BHSx · 2026-04-03
> >
> > Thanks for your response. It have fully resolved my concerns. I'll keep the current score.

---

> > > ### Author Response · Authors · 2026-04-07
> > >
> > > Thank you for your follow-up and for reviewing our response. We are glad that our response addressed your concerns.

---

### Decision · Program_Chairs · 2026-04-30

**Decision:**

Accept (regular)

**Comment:**

This paper studies the spectral anisotropy of gradient signals during LLM pretraining and argues that gradients exhibit a persistent low-rank-spike-plus-smooth-tail structure across scales, modules, and training stages. That empirical finding is itself a meaningful contribution: the paper does not merely introduce a new optimizer, but identifies a robust spectral pattern and uses it to motivate why Adam-style normalization and full-spectrum flattening can both be suboptimal. Building on this analysis, the paper proposes Spectra, a spike-aware optimizer that tracks only the dominant low-rank spike subspace via cached warm-started power iteration, shrinks the spike singular values toward the tail RMS scale, leaves the residual tail unchanged, and applies RMS-aligned updates. Experiments compare Spectra against AdamW, Muon, and Dion on Qwen3-0.6B and LLaMA3-8B pretraining, with additional gradient-spectrum analyses up to 32B scale and downstream evaluation.

**Strengths.** All four reviewers find this a strong paper: the topic is important, the writing is clear, and the optimizer story is coherent rather than ad hoc (BHSx, Fwmz, 9tL3, jJ3M). A major strength is that the paper does not simply propose a new optimizer and report better curves — it builds a consistent story from empirical spectral analysis to controlled interventions (FreqNorm, Shuffle) to algorithm design. The paper's central scientific value is not only that Spectra works, but that it isolates a concrete and practically consequential spectral phenomenon in LLM optimization. The method is practically appealing — tracking only a low-rank spike subspace is much lighter than full-spectrum processing — and experiments show a favourable trade-off between optimizer cost, wall-clock convergence, memory usage, and downstream accuracy, including strong LLaMA3-8B results. The rebuttal was effective and substantially raised confidence by clarifying implementation fairness, adding larger-scale runtime evidence up to 32B, and strengthening the robustness story for the spike-shrinking design.

**Weaknesses.** Three concerns remain, mostly about framing and theory rather than whether the optimizer works. First, the analysis is meaningful but not yet a full theoretical derivation of the final optimizer: Theorem 2.1 and the rebuttal's optimizer-agnostic extension provide a useful analytical lens on the spike-tail phenomenon, but the mapping from that analysis to the exact Spectra update rule remains partly heuristic rather than fully derived (9tL3, jJ3M). Second, the semantic interpretation of spike versus tail should be phrased more carefully: the controlled interventions provide strong correlational evidence but not a clean causal decomposition separating "common structure" from "rare semantics" (jJ3M). Third, some design choices remain heuristic — the default rank ratio, the shrink-to-tail-average rule, and the RMS calibration convention; the rebuttal's robustness results partially address this but do not turn these into principled choices (jJ3M, 9tL3). The distributed-training advantage at large scale is supported by complexity arguments plus added rebuttal evidence, but not by a full systems evaluation in the main paper (9tL3).

**Decision.** I recommend **accept**. All four reviewers support acceptance with post-rebuttal scores $5, 5, 4, 4$; two reviewers marked concerns fully resolved and the remaining reviewers stayed positive with only framing-level caveats. For the camera-ready the authors should (i) clearly separate scale-specific wall-clock claims and (ii) clearly distinguish the observed spectral structure from the paper's semantic interpretation and present the latter as a hypothesis supported by correlational evidence rather than a causal conclusion.